# CLUH granules coordinate translation of mitochondrial proteins with mTORC1 signaling and mitophagy

David Pla-Martín[1,2,†] (iD), Désirée Schatton[1,3,†], Janica L Wiederstein[1,3], Marie-Charlotte Marx[1,3], Salim Khiati[4], Marcus Krüger[1,3,5] & Elena I Rugarli[1,3,5,*] (iD)

## Abstract

Mitochondria house anabolic and catabolic processes that must be balanced and adjusted to meet cellular demands. The RNA-binding protein CLUH (clustered mitochondria homolog) binds mRNAs of nuclear-encoded mitochondrial proteins and is highly expressed in the liver, where it regulates metabolic plasticity. Here, we show that in primary hepatocytes, CLUH coalesces in specific ribonucleoprotein particles that define the translational fate of target mRNAs, such as *Pcx*, *Hadha*, and *Hmgcs2*, to match nutrient availability. Moreover, CLUH granules play signaling roles, by recruiting mTOR kinase and the RNA-binding proteins G3BP1 and G3BP2. Upon starvation, CLUH regulates translation of *Hmgcs2*, involved in ketogenesis, inhibits mTORC1 activation and mitochondrial anabolic pathways, and promotes mitochondrial turnover, thus allowing efficient reprograming of metabolic function. In the absence of CLUH, a mitophagy block causes mitochondrial clustering that is rescued by rapamycin treatment or depletion of G3BP1 and G3BP2. Our data demonstrate that metabolic adaptation of liver mitochondria to nutrient availability depends on a compartmentalized CLUH-dependent post-transcriptional mechanism that controls both mTORC1 and G3BP signaling and ensures survival.

**Keywords** CLUH; G3BP; mitochondria; mTORC1; RNA metabolism
**Subject Categories** Autophagy & Cell Death; RNA Biology; Signal Transduction
**The EMBO Journal (2020) 39: e102731**

## Introduction

Traditionally considered as the powerhouse of the cell, mitochondria contribute in several ways to cell and tissue metabolism, by producing biosynthetic intermediates, hosting catabolic reactions, and participating in signaling pathways (Chandel, 2014; Spinelli & Haigis, 2018). To adapt their metabolic function to cellular needs, mitochondria change shape, fuse or divide, interact with other organelles, and are replaced by balanced biogenesis and turnover (Eisner *et al*, 2018). When nutrients are abundant, the expression of a subset of mitochondrial proteins involved in oxidative phosphorylation (OXPHOS) and mitochondrial translation is promoted in a mTORC1-dependent manner to enable the production of ATP necessary for protein synthesis, which is an energetically costly process (Morita *et al*, 2013; Saxton & Sabatini, 2017). Turnover of actively respiring mitochondria via mitophagy ensures the maintenance of a healthy organellar population (Melser *et al*, 2013). Rewiring of mitochondrial metabolism is crucial to survive transitions from nutrient sufficiency to nutrient deprivation. During starvation, mitochondria are mainly catabolic organelles that use amino acids and lipids released by autophagy and convert them into ketone bodies and ATP to promote survival (Spinelli & Haigis, 2018). Inhibition of mTORC1 suppresses energy-consuming anabolic pathways and leads to mitochondrial hyperfusion as a mechanism to transiently protect mitochondria from autophagy and to suppress apoptotic cell death (Rambold *et al*, 2011; Morita *et al*, 2017). Prolonged starvation ultimately induces removal of mitochondria (Kristensen *et al*, 2008). Elucidating mechanisms that control the dynamic changes of mitochondrial metabolism and turnover to adapt them to energy needs is paramount to understand how organisms survive upon stress and starvation.

An important feature of a successful mitochondrial adaptive response is to be fast, flexible, and reversible. Coordination of post-transcriptional events by ribonucleoproteins (RNPs) plays a fundamental role in living systems to respond in a quick and dynamic manner to environmental signals and stress (Keene, 2007; Gehring *et al*, 2017). RNA-binding proteins (RBPs) can control each step of the mRNA life cycle, determining stability or degradation, localization, and translation efficiency of mRNAs (Hentze *et al*, 2018). RBPs

1 Institute for Genetics, University of Cologne, Cologne, Germany
2 Institute for Vegetative Physiology, University of Cologne, Cologne, Germany
3 Cologne Excellence Cluster on Cellular Stress Responses in Aging-Associated Diseases (CECAD), University of Cologne, Cologne, Germany
4 MitoLab Team, Institut MitoVasc, UMR CNRS 6015, INSERM U1083, Université d'Angers, Angers, France
5 Center for Molecular Medicine (CMMC), University of Cologne, Cologne, Germany
*Corresponding author. Tel: +49 221 47884244; E-mail: elena.rugarli@uni-koeln.de
†These authors contributed equally to this work

often assemble together with target transcripts in specific membrane-less subcellular compartments, such as stress granules (SGs), P-bodies, or other types of granules (Protter & Parker, 2016; Gomes & Shorter, 2019). These phase separations not only confer spatial regulation to the expression of groups of RNAs with a common function, but also integrate it with signaling pathways and allow sensing environmental changes (Kedersha *et al*, 2013; Yoo *et al*, 2019). Whether membrane-less organelles regulate mitochondrial function is currently unknown.

CLUH (clustered mitochondria homolog) is an RBP that specifically binds several transcripts encoding mitochondrial proteins (Gao *et al*, 2014; Schatton *et al*, 2017). At least for a subset of these, CLUH promotes their stability and translation (Schatton *et al*, 2017). Mitochondrial proteins whose expression depends on CLUH belong to several pathways, including OXPHOS, tricarboxylic acid (TCA) cycle, amino acid degradation, fatty acid oxidation, and ketogenesis (Schatton *et al*, 2017). In the absence of CLUH, the mitochondrial proteome is severely depleted of polypeptides encoded by mRNAs under CLUH regulation (Gao *et al*, 2014; Schatton *et al*, 2017). Mitochondria appear abnormal in ultrastructure and display a characteristic clustering next to the nucleus. This phenotype, which has given the name to the gene, is extremely conserved upon deletion of CLUH orthologues in evolutionary distant species (Fields *et al*, 1998, 2002; Logan *et al*, 2003; Cox & Spradling, 2009; Gao *et al*, 2014; Schatton *et al*, 2017). CLUH-deficient cells show metabolic abnormalities characterized by respiratory deficiency, a shift toward a glycolytic metabolism, and impairment of the TCA cycle and β-oxidation (Schatton *et al*, 2017; Wakim *et al*, 2017). *In vivo*, CLUH plays a key role to allow survival during the fetal to neonatal transition, which is characterized by acute starvation and a shift to OXPHOS metabolism (Schatton *et al*, 2017). In the adult liver, CLUH is required to reach maximal respiratory capacity under nutrient sufficiency, but also to produce ketone bodies upon starvation (Schatton *et al*, 2017).

Despite its crucial role for mitochondrial function in the liver, it is unclear whether CLUH is a general regulator of mitochondrial gene expression or whether it has a specific role during the metabolic switches in response to physiological nutrient fluctuations. Here, we show that in primary hepatocytes, CLUH assembles with its bound mRNAs in specific RNP particles that function not only as compartments that coordinate the translation of target mRNAs, but also as signaling hubs that control the dynamics of mTORC1 activation and modulate the function of other RBPs, such as Ras-GTPase-activating protein SH3 domain-binding proteins 1 and 2 (G3BPs). Through this mechanism, CLUH promotes turnover of mitochondria and metabolic rewiring. These data demonstrate a role of CLUH-dependent RNA granules in hepatocytes to coordinate mitochondrial catabolism and nutrient-sensing signaling pathways, thus ensuring survival upon starvation.

# Results

### CLUH and its target mRNAs form G3BP1-positive RNA granules

We previously showed that CLUH plays a physiological role in the adult mouse liver upon starvation to allow amino acid catabolism, to produce ketone bodies, and to maintain glucose levels (Schatton

*et al*, 2017). Intriguingly, CLUH subcellular localization in the mouse liver and in primary hepatocytes changed depending on the nutrient condition. CLUH displayed a cytosolic punctate localization in the liver of fed mice, but formed bigger aggregates in the tissue of mice subjected to food deprivation (Fig EV1A and B). Similarly, when hepatocytes were cultured in basal glucose-rich medium, CLUH decorated small cytosolic puncta and a few bigger foci. In contrast, incubation of hepatocytes for 2 h in HBSS, a low-glucose medium devoid of serum and amino acids, increased CLUH redistribution to bigger structures, often located in the perinuclear region (Fig EV1A and C).

These results raised the possibility of CLUH assembly together with client mRNAs in RNP particles, which play a regulatory role in response to nutrient availability. To detect whether known CLUH target mRNAs localize to these granules, we combined immunofluorescence with *in situ* hybridization in primary hepatocytes. We selected two target transcripts highly expressed in the liver, the expression of which is reduced in the absence of CLUH (Schatton *et al*, 2017): *Pcx* (encoding pyruvate carboxylase involved in the carboxylation of pyruvate to oxaloacetate) and *Hadha* (encoding hydroxyacyl-CoA dehydrogenase that catalyzes the last three steps of β-oxidation of long fatty acids). As a negative control, we analyzed the distribution of *Actb* mRNA. Under basal conditions, very little colocalization was observed between CLUH and each mRNA species (Fig 1A–F). However, we noticed that *Pcx* and *Hadha* mRNA molecules colocalized with CLUH only within the few granules present (Fig 1A and B). After HBSS starvation, the pattern of *Pcx* and *Hadha* mRNA molecules became visibly more aggregated, and the colocalization with CLUH significantly increased (Fig 1A, B, D and E). In contrast, colocalization of CLUH with *Actb* mRNA was not enhanced by starvation and remained at background levels (Fig 1C and F).

To further investigate the nature of the CLUH particles, we examined whether they contained markers of well-characterized RNP granules, such as SGs and P-bodies. SGs form under conditions of stress and contain mRNAs stalled in translation initiation together with specific RBPs (Panas *et al*, 2016; Protter & Parker, 2016), while P-bodies are constitutively present and contain translationally repressed mRNAs (Hubstenberger *et al*, 2017). Upon HBSS treatment, CLUH colocalized with TIA-1 and G3BP1, two RBPs that are present in SGs (Fig 1G and H). However, classical G3BP1-positive SGs induced by arsenite treatment did not contain CLUH (Fig EV1D). Furthermore, CLUH granules were not positive for DCP1A, a marker of P-bodies, although they were closely located (Fig EV1E).

In conclusion, CLUH forms granules with its target mRNAs in primary hepatocytes. These granules are positive for other RBPs and are more prominent upon starvation, but they are not induced by a classical SG triggering stimulus.

### CLUH-dependent granules temporally regulate translation of target mRNAs

G3BP1 and TIA-1 are markers for SGs where mRNAs are stalled after recruitment of the translation initiation complex (Panas *et al*, 2016; Protter & Parker, 2016). Therefore, our first hypothesis was that the assembly of CLUH and its target mRNAs in these granules reflected translational arrest upon nutrient stress. To investigate the

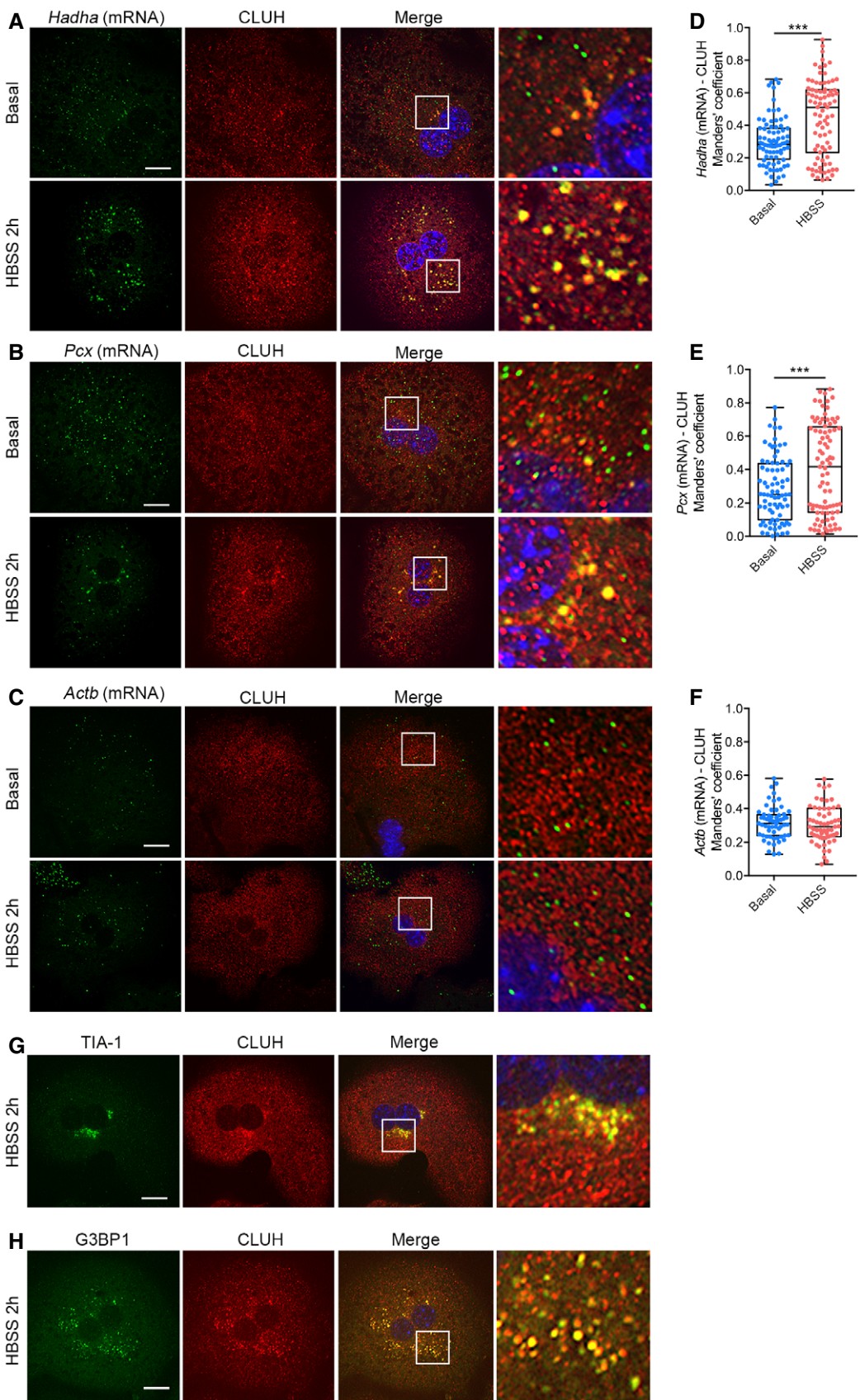

**Figure 1.**

◀

**Figure 1. CLUH forms specific RNA granules with its targets.**

A–C   Confocal images of primary hepatocytes grown under indicated conditions and stained with anti-CLUH antibody and *Hadha* (A), *Pcx* (B), and *Actb* (C) mRNA *in situ* hybridization. Right panels show 5× magnified boxed areas. Scale bar, 10 μm.

D–F   Manders' colocalization coefficient between *Hadha* (D), *Pcx* (E), and *Actb* (F) mRNA molecules and CLUH signal ($n \geq 50$ cells isolated from 3 to 6 mice).

G, H   Confocal images of primary hepatocytes grown in HBSS and stained with anti-CLUH and anti-TIA-1 (G) or anti-G3BP1 (H) antibodies. Right panels show 5× magnified boxed areas. Scale bar, 10 μm.

Data information: In (D–F), data are presented as boxplots showing the median, the first quartile, and the third quartile. Error bars show minimum and maximum values. ***$P \leq 0.001$ (Student's *t*-test).

translational status of the CLUH-positive granules, we made use of the ribopuromycylation assay (David *et al*, 2012), which reveals the subcellular localization of protein translation, by detecting the incorporation of puromycin into translating polypeptides with a specific antibody (Appendix Fig S1A). Hepatocytes cultured in basal medium showed a diffuse puromycin signal in the cytosol, indicating pervasive translation (Fig 2A and B). As expected, this signal was suppressed by pre-incubation with both homoharringtonine (HHT) and arsenite. HHT causes ribosome stalling at the initiator codon, but leaves unaffected downstream ribosomes already engaged in elongation, while arsenite is a potent inducer of SGs (Fig 2A and B). Upon HBSS treatment, the puromycin signal showed a more granular pattern (Fig 2A). While the total fluorescent intensity of puromycin signal was dramatically reduced by incubation with HHT (Fig 2B), the number of puromycin granules was reduced but not completely abolished by HHT, possibly indicating translation stalled at the level of elongation (Fig 2C). We found that colocalization of puromycin signal with CLUH was higher in HBSS than in basal medium and was significantly reduced by HHT treatment (Fig 2D and E, and Appendix Fig S1B). Furthermore, the percentage of puromycin-positive granules also showing CLUH signal was reduced when samples were pre-treated with HHT (Fig 2F). These data suggest the existence of two types of puromycin-positive compartments in HBSS, one harboring active translation (which disappears upon HHT treatment) and another where translation might be stalled. Both types of granules contain CLUH.

We combined *in situ* hybridization and the ribopuromycylation assay to correlate the translational state of the granules with specific CLUH mRNA targets. Remarkably, CLUH granules containing *Hadha* or *Pcx* mRNAs incorporated puromycin when cells were cultured in glucose, but were mostly puromycin-negative after HBSS incubation (Fig 3A, B, D and E, Appendix Fig S2A and B). We hypothesized that the starvation-induced CLUH granules are dynamic and regulate translation of mRNAs depending on cellular requirements. We therefore probed the *Hmgcs2* transcript, which encodes the mitochondrial enzyme 3-hydroxy-3-methylglutaryl-CoA synthase 2, implicated in the first reaction of ketogenesis. Defective production of β-hydroxybutyrate is the main metabolic defect of mice lacking CLUH specifically in the adult liver after starvation (Schatton *et al*, 2017). *Hmgcs2* mRNA molecules were detected both in basal and in HBSS conditions. However, the colocalization of CLUH and puromycin was prominent only in HBSS medium (Fig 3C and F; Appendix Fig S2C). To test whether these granules reflect stalled translation in HBSS, we performed control experiments with HHT. Pre-incubation with HHT completely abrogated the detection of *Hadha*, *Pcx*, or *Hmgcs2* mRNA molecules together with CLUH and puromycin (Appendix Fig S3A–C), demonstrating that CLUH granules are compartments where these mRNAs are translated.

## CLUH granules form in the absence of G3BPs and are distinct from SGs

To test whether CLUH overexpression triggers granule formation in the absence of any stress, we transfected an untagged version in HeLa cells and examined overexpressing versus non-overexpressing cells in the same dish, using a specific antibody (Figs 4A and B, and EV2A and B). CLUH overexpression induced the formation of peripheral granules, positive for G3BP1 and the homolog protein G3BP2, in approximately 40% of transfected cells (Figs 4B and C, and EV2C and D). However, CLUH granules were not abrogated by cycloheximide (CHX) treatment, in contrast to arsenite-induced G3BP1-granules, indicating that they are not SGs (Fig EV2E–H). We downregulated G3BP1, G3BP2, or both RBPs and examined whether overexpressed CLUH retained the ability to induce granules (Fig 4A–C). The formation of CLUH granules was affected neither by G3BP1 downregulation nor by concomitant downregulation of both G3BPs (Fig 4B and C). Significantly less CLUH granules were seen in cells depleted for G3BP2, although the levels of overexpressed CLUH were decreased (Fig 4A). Thus, CLUH overexpression is sufficient for granule formation, independently from G3BP1 and G3BP2.

We then asked whether G3BP1 granules can still form in HeLa cells lacking CLUH (Wakim *et al*, 2017). To this purpose, we expressed G3BP1-GFP and used live imaging to follow the formation of granules upon HBSS incubation. We selected cells which did not show granules at the beginning of the imaging, to avoid analysis of SGs that are known to form upon G3BP1 overexpression (Protter & Parker, 2016). Strikingly, approximately 80% of WT cells formed G3BP1-positive granules in HBSS, while this rarely occurred in KO cells (Fig 4D and E). These granules were still formed upon CHX treatment, suggesting that most of them are not SGs (Fig 4D and E). In contrast, arsenite treatment induced classical CHX-sensitive SGs in both WT and KO cells (Fig EV2I and J). Therefore, CLUH is required for the efficient formation of starvation-induced G3BP1 granules in HeLa cells.

Lastly, we obtained hepatocytes from liver-specific *Cluh* knock-out (KO) mice (Li-*Cluh*[KO]) (Schatton *et al*, 2017) and used G3BP1 staining to detect granules. The number of cells showing G3BP1-positive granules was significantly lower in CLUH-deficient hepatocytes compared to control cells in both basal and HBSS conditions (Fig 4F and G). Intriguingly, the protein levels of G3BP1 were also lower in KO hepatocytes under basal conditions, but they increased similarly in HBSS in cells of both genotypes (Fig 4H and I). In contrast, G3BP1 mRNA levels were unaffected by lack of CLUH (Fig 4J). Thus, our data suggest that CLUH regulates G3BP1 abundance, independent from transcriptional regulation.

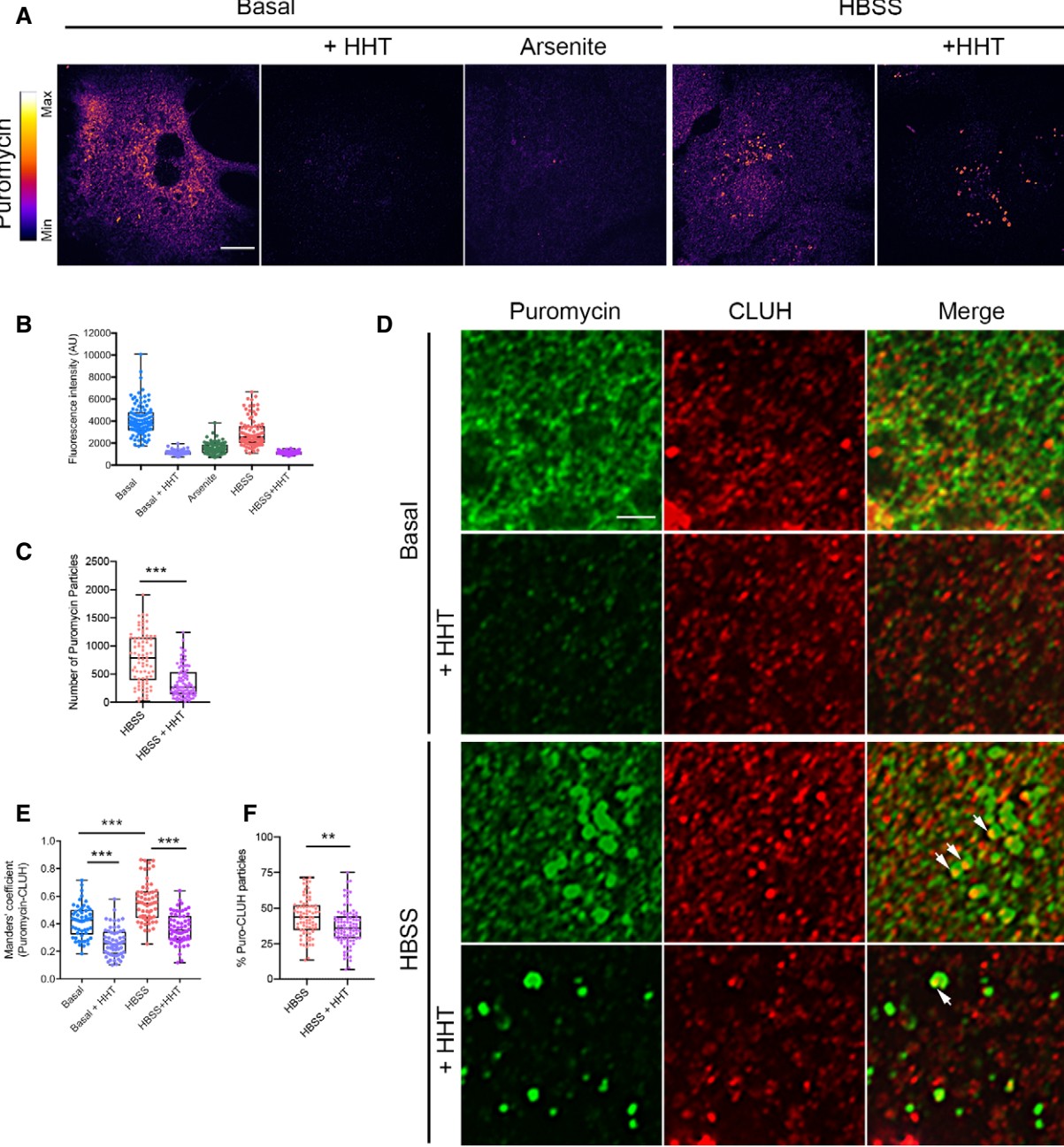

**Figure 2. CLUH granules contain stalled and active translation sites.**

A   Confocal images of primary hepatocytes grown under indicated conditions and treatments after ribopuromycylation assay stained with anti-puromycin antibody. Scale bar, 10 μm.

B   Quantification of fluorescence intensity per cell of experiment shown in (A). AU, arbitrary units (*n* = 90–110 cells per treatment isolated from 4 mice).

C   Quantification of the number of puromycin granules under indicated conditions (*n* ≥ 50 cells isolated from 4 mice).

D   Confocal images of primary hepatocytes stained with anti-puromycin and anti-CLUH antibodies. The cells from which the enlarged areas (400 μm²) have been magnified are shown in Appendix Fig S1B for each individual channel. Scale bar, 4 μm. Cells analyzed were isolated from 4 different mice with similar results. Arrows point to colocalizing particles.

E   Manders' colocalization coefficient between puromycin and CLUH from experiment shown in (D) (*n* ≥ 50 cells isolated from 4 mice).

F   Quantification of puromycin granules containing CLUH signal (*n* ≥ 80 cells isolated from 4 mice).

Data information: In (B, C, E, F), data are presented as boxplots showing the median, the first quartile, and the third quartile. Error bars show minimum and maximum values. (C, F) ***$P$ ≤ 0.001; **$P$ ≤ 0.01 (Student's *t*-test). (E) ***$P$ ≤ 0.001 (one-way ANOVA, Tukey's multiple comparison test).

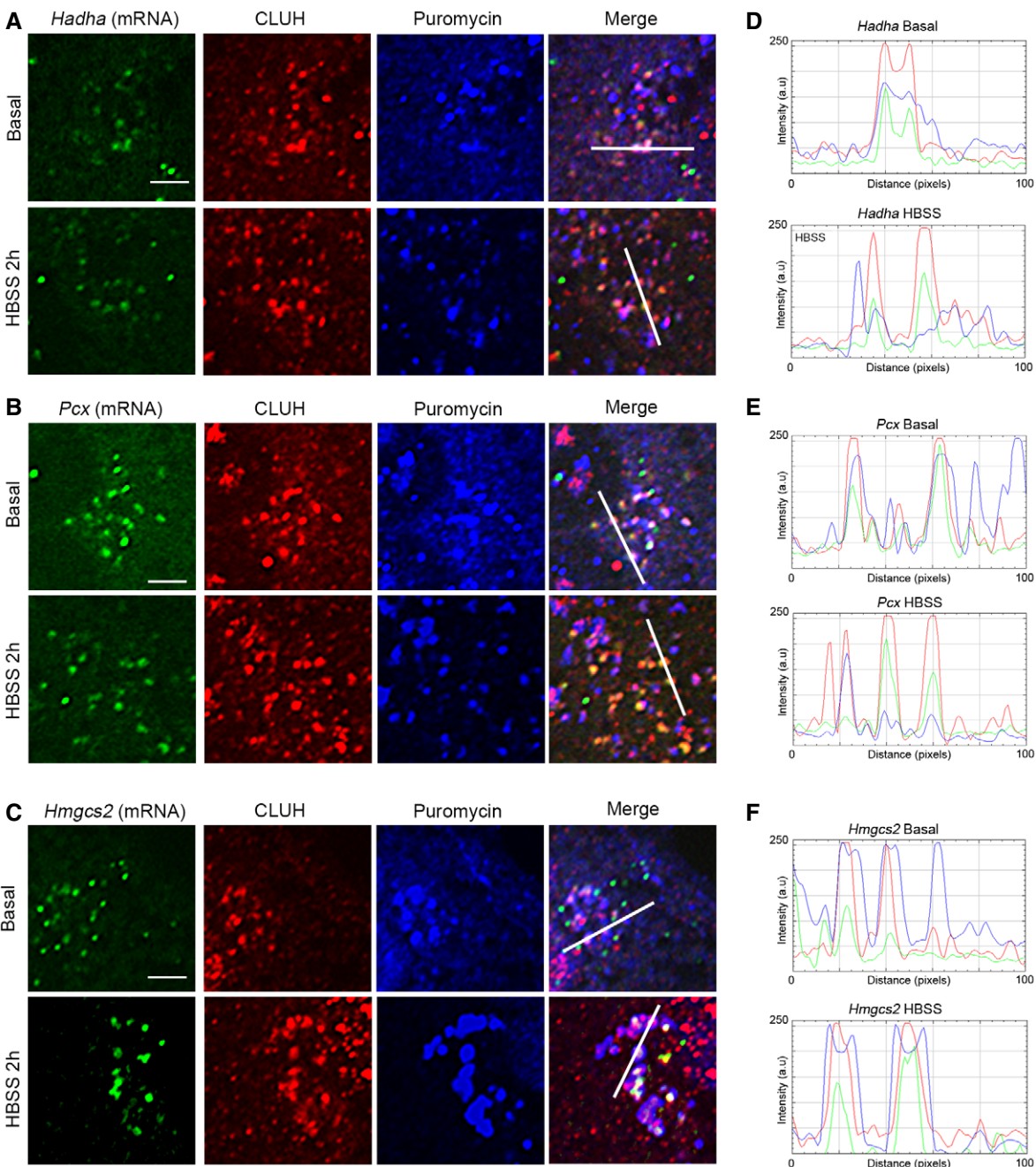

**Figure 3. CLUH granules are translationally active or dormant depending on the mRNA.**

A–C  Confocal images of primary hepatocytes after ribopuromycylation experiment combined with mRNA *in situ* hybridization for (A) *Hadha*, (B) *Pcx*, and (C) *Hmgcs2*. Scale bar, 4 μm.

D–F  Fluorescence profile of 100-pixel line from the merged channel shown in (A–C). The cells from which the enlarged areas (400 μm²) have been magnified are shown in Appendix Fig S2 for each individual channel. Cells analyzed were isolated from 6 different mice with similar results.

## CLUH differentially regulates mRNAs involved in mitochondrial anabolic and catabolic pathways

To further understand the physiological role of CLUH granules, we performed RNA-seq and quantitative label-free mass spectrometry (MS) on wild-type (WT) and *Cluh*-deficient hepatocytes (Dataset EV1 and EV2). To systematically detect groups of mRNAs that participate in similar pathways and to reveal early proteome changes after the short starvation period, we performed one-dimensional (1D) pathway enrichment of transcriptome and proteome data (Dataset EV3), which allows the identification of even small shifts in pathways by summarizing the fold changes of all proteins

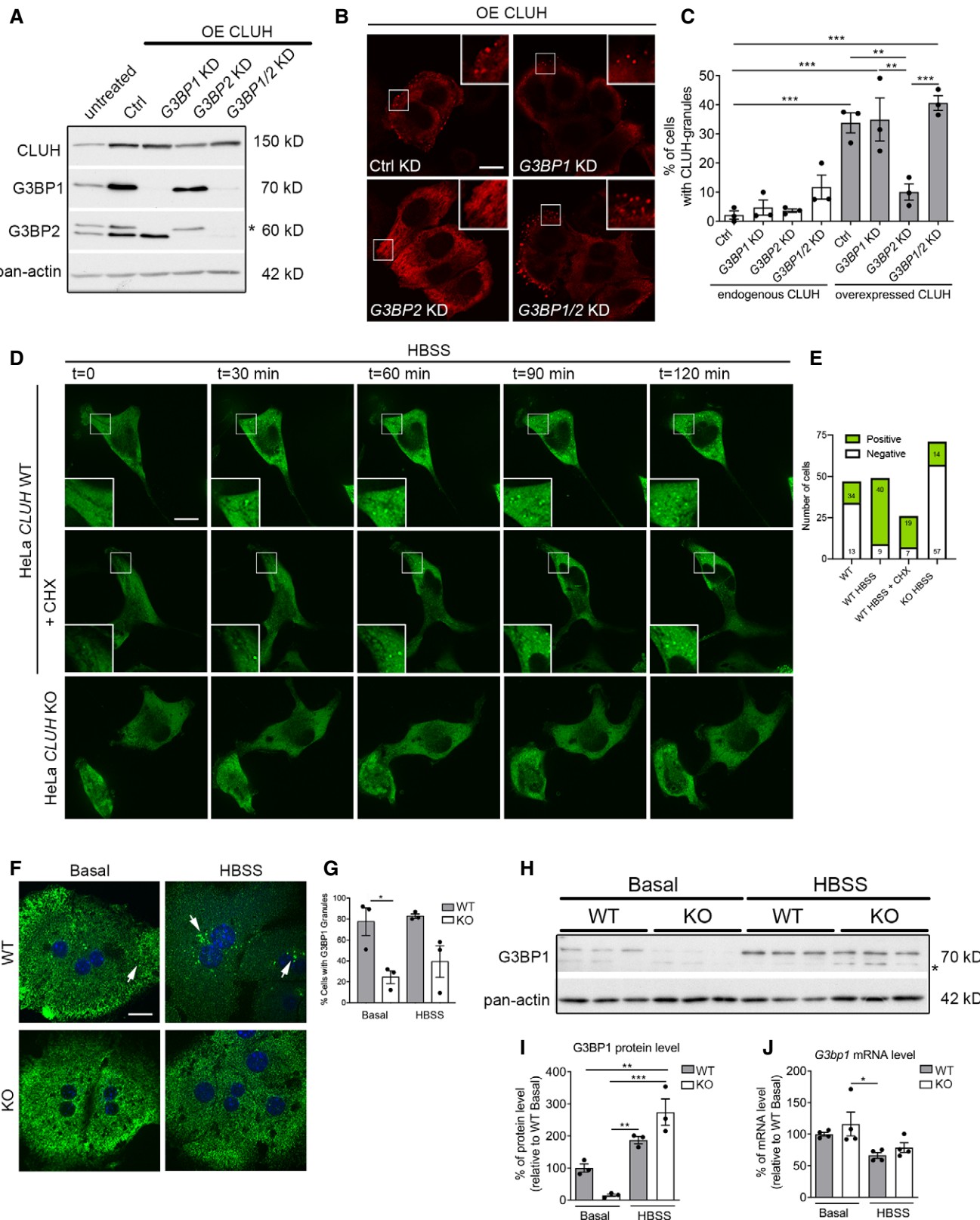

**Figure 4.**

Figure 4.  CLUH granules form in the absence of G3BPs and are distinct from SGs.

A   Representative Western blot of HeLa cells downregulated for G3BPs and overexpressing untagged CLUH. Asterisks indicate signal of previous incubation with anti-G3BP1 antibody. Pan-actin was used as loading control.
B   Confocal images of anti-CLUH staining in HeLa cells downregulated for G3BPs and overexpressing untagged CLUH. Insets show 2.6× enlarged area. Scale bar, 10 μm.
C   Quantification of cells with CLUH granules from experiments shown in (B) and Fig EV2B (*n* = 3 independent experiments, > 50 cells per experiment per condition).
D   Live imaging of WT and *CLUH* KO HeLa cells transfected with G3BP1-GFP plasmid and incubated in HBSS medium with or without CHX. Cells were recorded for a maximum of 2 h. Insets show 2.5× enlarged areas. Scale bar, 10 μm.
E   Total number of cells analyzed by live imaging for the indicated experiments shown in (D). "Positive" relates to a cell which formed G3BP1 granules at the end of the recording.
F   Confocal images of anti-G3BP1 staining in primary hepatocytes derived from Li-*Cluh*[WT] and Li-*Cluh*[KO] mice grown in indicated media. Arrows point to G3BP1-granules. Scale bar, 10 μm.
G   Quantification of percentage of cells with G3BP1-positive granules of experiment shown in (F). Cells were analyzed in a blind fashion (*n* = 3 mice per genotype; number of cells in total: WT basal, 449; KO basal, 303; WT HBSS, 146; KO HBSS, 161).
H   Western blot of primary WT and KO hepatocytes stained with indicated antibodies. Asterisks indicate unspecific signal. Pan-actin was used as loading control.
I   Quantification of Western blot shown in (H) (*n* = 3 per genotype per condition).
J   mRNA levels of *G3bp1* in primary hepatocytes grown under indicated conditions (*n* = 4 per genotype per condition).

Data information: In (C, G, I, J), data are presented as histograms showing the mean ± SEM. *$P \leq 0.05$; **$P \leq 0.01$; ***$P \leq 0.001$ (one-way ANOVA, Tukey's multiple comparisons test).
Source data are available online for this figure.

annotated with a specific term. Differently regulated terms between conditions were identified based on a false discovery rate, FDR, < 0.02.

To examine whether WT and KO cells respond in a similar manner to starvation, we first examined changes upon shifting WT or KO hepatocytes from basal glucose-rich medium to HBSS for 2 h. Starvation triggered a modest reshaping of the transcriptome and proteome in cells of both genotypes; however, analysis of pathways enriched or depleted specifically in WT or KO cells revealed differences (Appendix Fig S4A and B, and Dataset EV3). Terms associated with mitochondrial translation and mitochondrial respiratory complex I were enriched in transcriptomics of starved *Cluh* KO hepatocytes (Appendix Fig S4B and Dataset EV3), but not detected in starved control cells. At the proteome level, proteins involved in biosynthetic fatty acid processes were less represented in control cells, as expected upon starvation, while proteins involved in actin dynamics and ribonucleoprotein complexes were increased (Appendix Fig S4C and Dataset EV3). Instead, several terms linked with cap-dependent translation initiation were decreased in the absence of CLUH upon starvation, while terms related to mRNA decay and lysosomes were enriched (Appendix Fig S4D and Dataset EV3). These results suggest that WT and KO cells respond differently to starvation, and are consistent with a role of CLUH to protect target mRNAs from degradation and to promote a translation response under conditions of nutrient deprivation.

We then compared mRNA and protein changes between WT and KO hepatocytes (Fig 5A–D and Dataset EV1 and EV2). A subset of mRNAs and the corresponding mitochondrial proteins (Fig 5A and B) were significantly decreased in abundance. One-dimensional enrichment analysis of the data showed a global under-representation at both mRNA and protein level of several mitochondrial pathways, such as pyruvate metabolism, β-oxidation, and amino acid degradation, in both basal conditions and HBSS in CLUH-deficient hepatocytes (Fig 5C–E, blue, Dataset EV3), in agreement with our previous findings in embryonic livers (Schatton *et al*, 2017). However, pathways associated with mitochondrial translation and the mitochondrial large ribosomal subunit were enriched on the proteome level (Fig 5D, red, Dataset EV3), but decreased on the transcriptome level (Fig 5E, red, Dataset EV3) when CLUH was

absent. We then compared the protein fold changes between WT and KO specifically for those proteins whose mRNA was previously identified in CLUH immunoprecipitations in HeLa cells (Gao *et al*, 2014). As expected, in KO hepatocytes the majority of these proteins showed a significant downregulation (Fig 5F). However, proteins encoded by a subset of target mRNAs were not affected or slightly upregulated (Fig 5F, inset box). A closer inspection of these proteins revealed that they comprised several mitochondrial ribosomal proteins and the mitochondrial transcription factor TFAM. We conclude that CLUH positively regulates mRNA and protein levels of mRNAs involved in catabolic pathways, such as amino acid degradation, fatty acid oxidation, and the TCA cycle, but may have a translationally inhibitory effect on another group of transcripts, involved in mitochondrial transcription and translation.

### CLUH ensures mTORC1 inhibition upon starvation

The expression of TFAM and of mitochondrial ribosomal proteins is known to be regulated at the translational level by mTORC1, specifically via inhibition of the eukaryotic translation initiation factor 4E-binding proteins (4E-BPs), to sustain ATP production during growth (Thoreen *et al*, 2012; Morita *et al*, 2013). Therefore, we investigated the level of mTORC1 activation in primary hepatocytes during HBSS starvation, by assessing the phosphorylation of downstream substrates. The phosphorylation of 4E-BP1 was higher in KO hepatocytes in both basal conditions and at different time points after HBSS starvation, while the phosphorylation of ribosomal protein S6 (RPS6) was not significantly increased (Fig 6A–C). We also examined whether Li-*Cluh*[KO] mice could efficiently suppress mTORC1 activation during starvation. Similar to hepatocytes, the levels of phosphorylated 4E-BPs were higher in the KO livers after 24-h starvation than in controls (Fig 6D and E). The increase in the amount of phosphorylated RPS6 was less consistently observed (Fig 6D and F).

mTORC1 suppression during nutrient deprivation is fundamental to protect cells from apoptosis. One of the most upregulated proteins in KO hepatocytes compared to control cells in starvation was the pro-apoptotic protein Diablo (Fig 5B). In agreement with this result, we found increased caspase-3 cleavage in CLUH-deficient

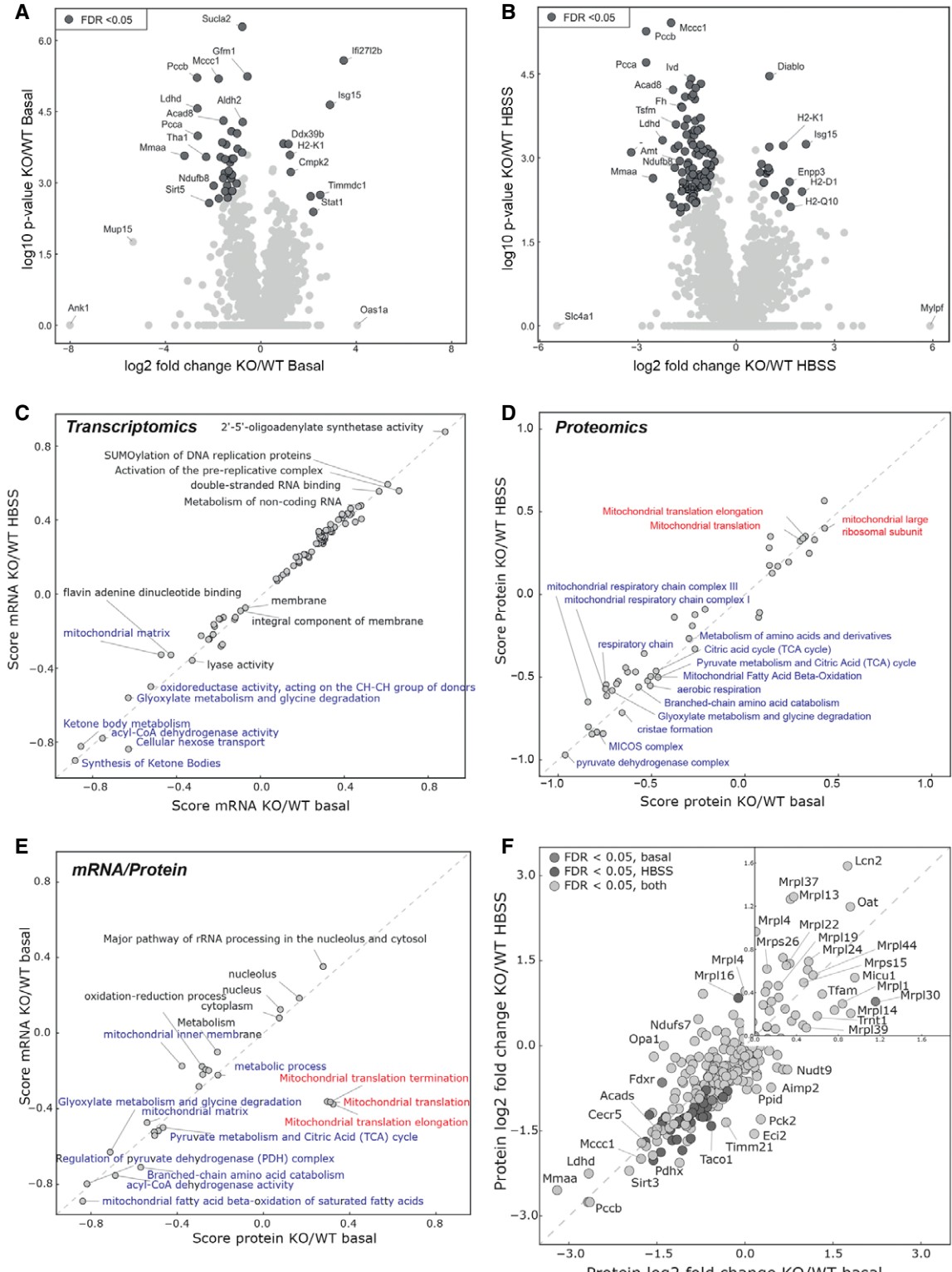

**Figure 5. Transcriptomic and proteomic profile of WT and KO hepatocytes.**

A, B   Volcano plots showing significantly changed proteins in KO hepatocytes relative to WT in basal (A) and HBSS (B) conditions.

C   2D score plots of enriched pathways in transcriptomics analysis of KO and WT hepatocytes under basal condition and upon HBSS (2 h).

D   2D score plots of enriched pathways in proteomics analysis of KO and WT hepatocytes under basal condition and upon HBSS (2 h).

E   2D score plots of enriched pathways in proteomics versus transcriptomics analysis of KO and WT hepatocytes under basal condition.

F   Correlation of protein fold changes in KO with respect to WT hepatocytes under basal condition and upon HBSS (2 h). Only genes previously found in CLUH RIP experiments in HeLa cells (Gao *et al*, 2014) are plotted. Inset shows magnification of upper part of the graph.

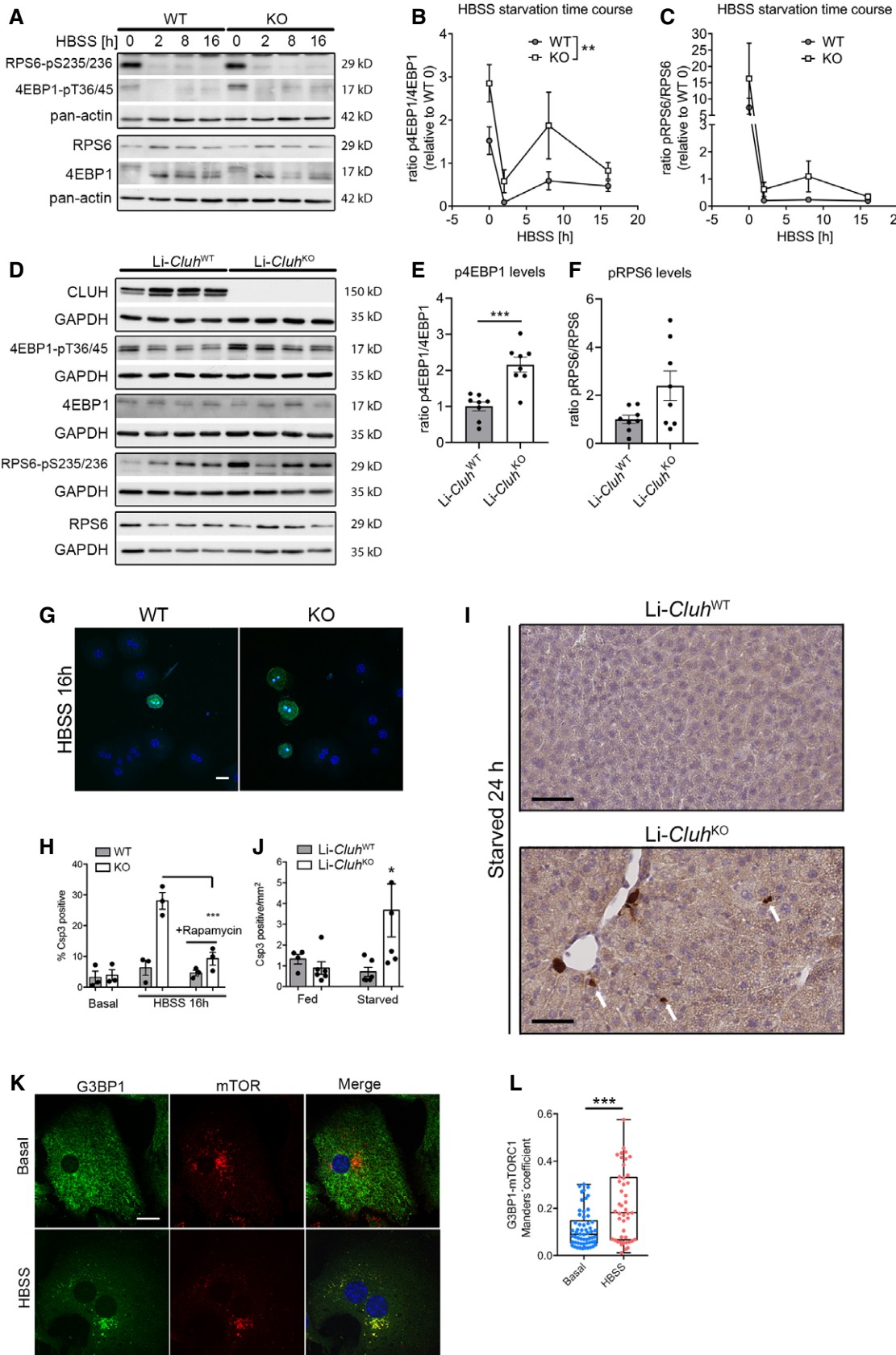

**Figure 6.**

**Figure 6.  CLUH regulates mTORC1 activation and inhibits apoptosis upon starvation.**

A       Representative Western blot of primary hepatocytes starved in HBSS for the indicated time points and probed with indicated antibodies. Pan-actin was used as loading control.
B, C    Quantification of Western blots as shown in (A). Antibody signal was normalized to pan-actin signal, and signal of phospho-protein was normalized to signal of total protein (*n* = 4 independent experiments).
D       Representative Western blot of total protein extracts from livers of starved Li-*Cluh*^WT and Li-*Cluh*^KO mice probed with indicated antibodies. GAPDH was used as loading control.
E, F    Quantification of Western blots as shown in (D). Antibody signal was normalized to GAPDH signal, and signal of phospho-protein was normalized to signal of the total protein (*n* = 8 mice per genotype).
G       Confocal images of primary hepatocytes grown in HBSS for 16 h and stained with anti-cleaved caspase 3 antibody. Scale bar, 25 μm.
H       Quantification of cleaved caspase 3-positive hepatocytes treated with or without rapamycin. More than 100 cells per replicate were analyzed in a blind fashion (*n* = 3 independent experiments).
I       IHC staining of cleaved caspase 3 in liver sections of starved Li-*Cluh*^WT and Li-*Cluh*^KO mice. Arrows point to positive apoptotic foci. Scale bar, 200 μm.
J       Quantification of positive apoptotic foci of pictures shown in (I). The number of positive spots was relativized to the area analyzed (*n* = 5 mice per genotype).
K       Confocal images of primary hepatocytes grown in indicated media and stained with anti-G3BP1 and anti-mTOR. Scale bar, 10 μm.
L       Manders' colocalization coefficient between G3BP1 and mTOR staining (*n* > 50 cells isolated from 3 mice).

Data information: In (B, C), data show the mean ± SEM. **$P \leq 0.01$ for genotype and ***$P \leq 0.001$ for time (two-way ANOVA). In (E, F, H, J), data are presented as histograms showing the mean ± SEM. In (L), data are presented as boxplots showing the median, the first quartile, and the third quartile. Error bars show minimum and maximum values. (E, F, L) ***$P \leq 0.001$ (Student's *t*-test). (H, J) *$P \leq 0.05$; ***$P \leq 0.001$ (one-way ANOVA, Tukey's multiple comparison test).
Source data are available online for this figure.

hepatocytes cultured for 16 h in HBSS, which was rescued by treatment with the allosteric mTORC1 inhibitor rapamycin (Fig 6G and H). A higher number of apoptotic cells were also detected in the liver of Li-*Cluh*^KO mice upon starvation (Fig 6I and J).

mTORC1 coordinates an integrated stress response in the presence of mitochondrial dysfunction (Khan *et al*, 2017), potentially explaining our findings. However, neither a significant change in the expression of the transcription factors *Atf4* and *Atf5*, nor upregulation of the mRNA levels of *Psat1*, *Phgdh*, or *Mthfd2*, all markers of the mitochondrial integrated stress response, were observed in KO hepatocytes or livers (Appendix Fig S5A and B). mTORC1 can also be inhibited by dynamic recruitment of its components to RNA granules, for example, to SGs upon stress (Takahara & Maeda, 2012; Wippich *et al*, 2013), raising the possibility that CLUH granules may directly regulate the extent of the mTORC1 signaling response upon starvation. In agreement with this hypothesis, the G3BP1-positive granules which form in starved primary hepatocytes contained the mTOR kinase (Fig 6K and L).

These data highlight a crucial role of CLUH granules not only to coordinate the translation of nuclear-encoded mitochondrial proteins involved in amino acid and fatty acid catabolism and ketogenesis, but also to suppress the anabolic role of mTORC1 upon starvation, thereby protecting from cell death.

**Lack of CLUH is associated with increased bulk autophagy, but decreased mitophagy**

Our data indicate that CLUH granules may play signaling functions, by recruiting the kinase mTOR and other RBPs, like G3BP1 and G3BP2. mTORC1 controls autophagy at different steps: It inhibits not only the initiation of autophagy, but also late steps of autophagy, such as the fusion of autophagosomes and lysosomes, lysosomal biogenesis, and reformation (Yu *et al*, 2010; Zhou *et al*, 2013; Shen & Mizushima, 2014). G3BP1 and G3BP2 are well known for their role in SG formation; however, they play a plethora of other signaling functions (Alam & Kennedy, 2019). Of note, the yeast orthologue of G3BPs, Bre5, has been shown to promote ribophagy and other types of autophagy, and to inhibit mitophagy (Muller

*et al*, 2015). Our data open up the possibility that CLUH RNP particles regulate autophagic pathways, which are essential to survive starvation. To analyze the impact of CLUH deficiency on general autophagy in primary hepatocytes and in the liver, we crossed Li-*Cluh*^KO mice with transgenic mice expressing the LC3-GFP reporter (Mizushima *et al*, 2004) and analyzed them 4 weeks after complete CLUH deficiency was achieved (Schatton *et al*, 2017). In the absence of CLUH, LC3-GFP accumulated in larger aggregates in the liver and in primary hepatocytes (Fig 7A–E), indicating that autophagosomal formation was not impaired upon sustained CLUH deficiency. In addition, CLUH-deficient livers showed increased amount of LAMP1, a marker of lysosomes (Fig 7F and G). These results can reflect an increased autophagic flux, or a block in the consumption of autophagosomes. To distinguish between these possibilities, we analyzed the autophagic flux in primary hepatocytes, by measuring the conversion of LC3-I to LC3-II in the absence or presence of bafilomycin A, a potent inhibitor of the vacuolar H$^+$ ATPase. Surprisingly, lack of CLUH led to an increase in the autophagic flux under basal conditions (Fig 7H–J). The autophagic flux was increased in both WT and KO cells upon HBSS, as indicated by consumption of the autophagic adaptor p62 (Fig 7H and K). Interestingly, the steady-state levels of CLUH were significantly reduced upon starvation (Fig 7L). Together, our results indicate that autophagy is increased in hepatocytes lacking CLUH already under basal conditions, despite mTORC1 hyperactivation.

Granulophagy is a specialized form of autophagy that removes RNP particles, including SGs (Frankel *et al*, 2017). We hypothesized that the decrease in G3BP1 levels and G3BP1-positive granules in KO hepatocytes already in basal conditions could be caused by their increased removal by autophagy. We analyzed this possibility by assessing the colocalization of G3BP1 granules with lysosomes in basal conditions and after a short (30 min) or more prolonged (2 h) starvation in HBSS. In WT hepatocytes, G3BP1 granules showed colocalization with LAMP1 after 2 h of starvation, while when G3BP1 granules were observed in KO hepatocytes, they colocalized with LAMP1 after only 30 min in HBSS medium (Fig EV3A–C). We conclude that CLUH protects G3BP1-positive granules from premature autophagic degradation upon starvation.

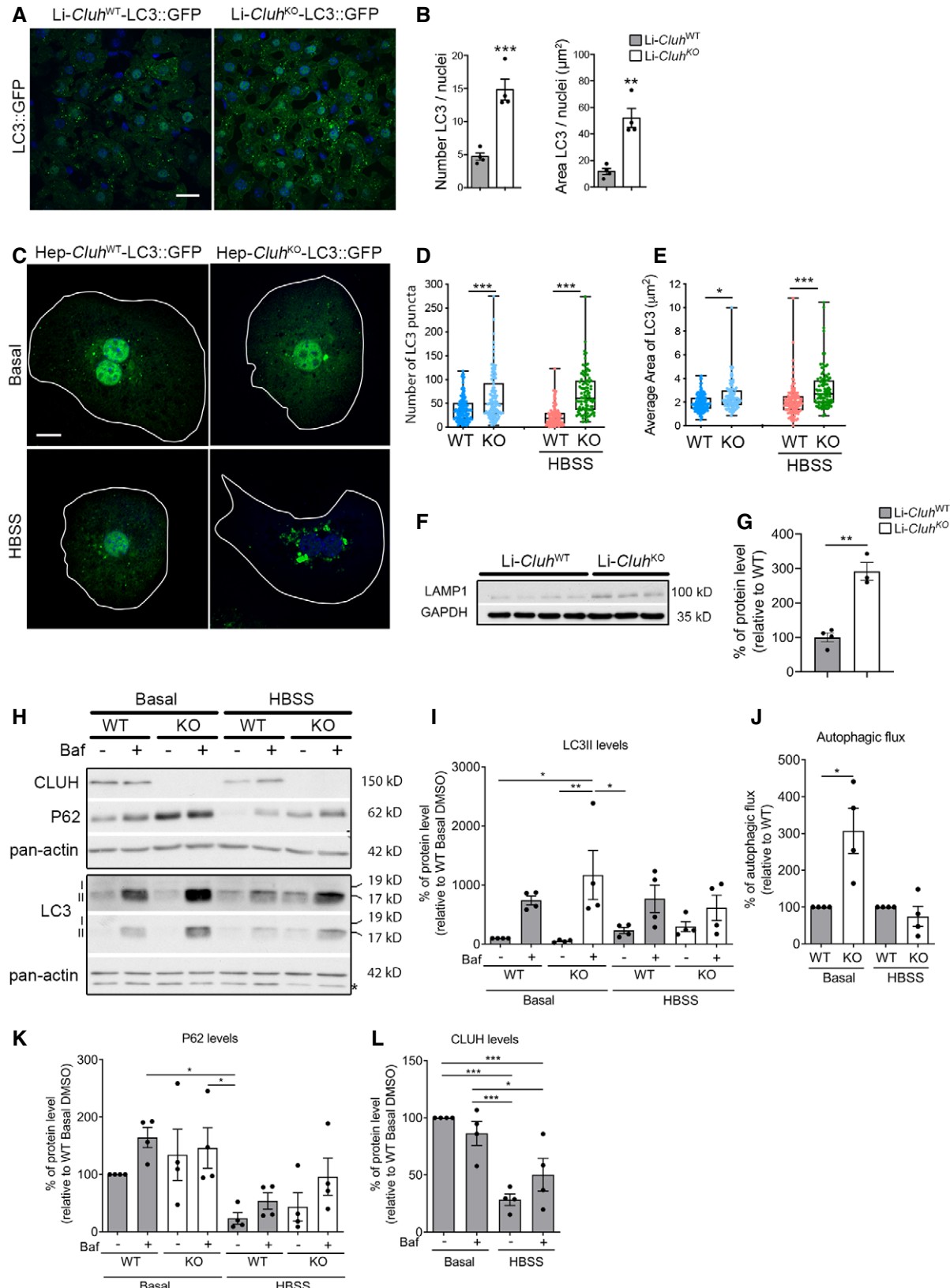

Figure 7.

**Figure 7.  Lack of CLUH promotes bulk autophagy.**

A       Confocal images of liver cryosections from Li-*Cluh*^WT and Li-*Cluh*^KO-LC3-GFP mice. Scale bar, 25 μm.
B       Quantification of LC3 signal morphology of images shown in (A). Graphs show relative number of LC3 autophagosomes and the relative area occupied per nuclei.
         Four images were quantified per animal and averaged. Nuclear staining was excluded for the analysis (*n* = 4 mice per genotype).
C       Confocal images of primary hepatocytes derived from Li-*Cluh*^WT and Li-*Cluh*^KO-LC3-GFP mice. Hepatocytes were starved for 2 h in HBSS. Scale bar, 10 μm.
D, E    Quantification of LC3 puncta number (D) and area (E) per cell (*n* ≥ 100 cells isolated from 3 mice).
F       Western blot of total protein extracts from livers of Li-*Cluh*^WT and Li-*Cluh*^KO mice probed with LAMP1. GAPDH was used as loading control.
G       Quantification of Western blot shown in (F). WT: *n* = 4; KO: *n* = 3.
H       Representative Western blot of primary hepatocytes treated for 2 h with bafilomycin A in the indicated media and probed with indicated antibodies. Two different
         exposure times are shown for LC3. I refers to total LC3, and II indicates lipidated form of LC3. Asterisk indicates signal of previous incubation.
I       Quantification of LC3 levels in Western blots as shown in (H) (*n* = 4 independent experiments).
J       Quantification of autophagic flux of Western blot as shown in (H). The flux was calculated as the increase in LC3II upon bafilomycin A treatment in each condition.
         The flux in KO cells was normalized to flux in WT cells (*n* = 4 independent experiments).
K, L    Quantification of p62 and CLUH levels in Western blots as shown in (H) (*n* = 4 independent experiments).

Data information: In (B, G, I–L), data are presented as histograms showing the mean ± SEM. In (D, E), data are presented as boxplots showing the median, the first
quartile, and the third quartile. Error bars show minimum and maximum values. (B, G, J) *$P$ ≤ 0.05; **$P$ ≤ 0.01; ***$P$ ≤ 0.001 (Student's *t*-test). (D, E, I, K, L) *$P$ ≤ 0.05;
**$P$ ≤ 0.01; ***$P$ ≤ 0.001 (one-way ANOVA, Tukey's multiple comparison test).
Source data are available online for this figure.

Surprisingly, p62 accumulated in CLUH-deficient livers (Fig 8A and B), despite functional autophagy. This prompted us to analyze mitophagy, a specialized form of autophagy that allows turnover of old or damaged mitochondria. We found that p62 and ubiquitin accumulated in liver mitochondrial lysates in the absence of CLUH (Fig 8C and D). Consistent with our previous findings, mitochondrial proteins encoded by CLUH target mRNAs analyzed in this study were decreased in purified mitochondria (Fig 8C and D). In further agreement with a mitophagy block, LC3 signal colocalized more with mitochondria in livers of Li-*Cluh*^KO mice compared to controls (Fig 8E and F). Furthermore, more LC3-positive puncta colocalized with mitochondria in KO hepatocytes, and rapamycin treatment was effective in restoring this colocalization to WT levels (Fig 8G and H). To corroborate these findings, we transfected WT and *Cluh*-deficient mouse embryonic fibroblasts (MEFs) with a construct expressing a tandem mCherry-GFP fluorescent protein targeted to mitochondria (mCherry-GFP-FIS$_{101-152}$). Upon engulfment of mitochondria in lysosomes, the GFP signal is quenched, while the red fluorescence is preserved, allowing it to distinguish and quantify mitochondria that have been successfully delivered to functional lysosomes (Allen *et al*, 2013). Mitophagy was induced by culturing cells in galactose, or by adding antimycin A and oligomycin (A/O) to the medium (Allen *et al*, 2013; Melser *et al*, 2013). Remarkably, turnover of mitochondria was impaired in cells depleted of CLUH in both conditions (Fig 8I and J) and rescued by rapamycin treatment (Fig 8K and L). Thus, constitutive lack of CLUH accelerates autophagy but inhibits mitophagy.

**Rapamycin treatment and G3BP downregulation rescue the mitochondrial clustering phenotype caused by lack of CLUH**

Since mitochondria cluster next to the microtubule-organizing center before being removed by mitophagy (Lee *et al*, 2010), the combination of dysfunctional mitochondria targeted for degradation with a mitophagy block would explain the evolutionary conserved phenotype of CLUH mutants (Zhu *et al*, 1997; Fields *et al*, 1998; Logan *et al*, 2003; Cox & Spradling, 2009; Gao *et al*, 2014; Sen *et al*, 2015; Wang *et al*, 2016). We predicted that restoring perturbed signaling responses caused by the lack of CLUH would improve mitochondrial clustering. Induction of autophagy by mTORC1

inhibition via daily injections of rapamycin (7.5 mg/kg for seven consecutive days) reduced mitochondrial aggregation and improved mitochondrial morphology parameters of Li-*Cluh*^KO mice (Fig 9A and B). We also noted that the lysosomal accumulation of Li-*Cluh*^KO mice was reverted by rapamycin treatment to the levels observed in mock-treated control mice (Fig 9C and D). In agreement with these results, treatment with rapamycin or torin, a selective and ATP-competitive mTORC1 inhibitor, prevented the prominent mitochondrial clustering observed in COS-7 cells acutely downregulated for CLUH (Fig EV4). Finally, we examined whether the mitochondrial clustering can be rescued by downregulating G3BPs (Fig 9E–G). As previously reported, depletion of G3BP1 led to an increase of G3BP2 levels, but not *vice versa* (Kedersha *et al*, 2016; Fig 9F). Depletion of each or both G3BPs did not cause substantial alterations in the mitochondrial network (Fig 9G). Strikingly, downregulation of G3BP1, G3BP2, or both effectively suppressed mitochondrial clustering observed in the absence of CLUH (Fig 9E and G).

Our data indicate that mitochondrial clustering induced by CLUH deficiency is secondary to altered signaling responses, involving both mTORC1 and G3BP proteins, which impair the efficient turnover of dysfunctional mitochondria.

## Discussion

We reveal a CLUH-dependent post-transcriptional mechanism that coordinates the mitochondrial catabolic response during starvation with signaling pathways that control bulk autophagy and the turnover of mitochondria. Central to this response is the formation of CLUH RNP particles, which act as spatial compartments to preserve mRNAs involved in mitochondrial catabolic pathways and allow their translation, but also as signaling hubs by recruiting the mTOR kinase and other RBPs, such as G3BP1 and G3BP2. Thus, CLUH plays a fundamental role to control the metabolic plasticity of hepatic mitochondria.

RNA granules are large, non-membrane-bound structures that reflect the coalescence of RBPs together with mRNAs via a phase transition from a soluble to a liquid droplet (Hyman *et al*, 2014). Several types of RNA granules, such as SGs, P-bodies, or neuronal RNA granules, have been shown to coordinately regulate the fate of

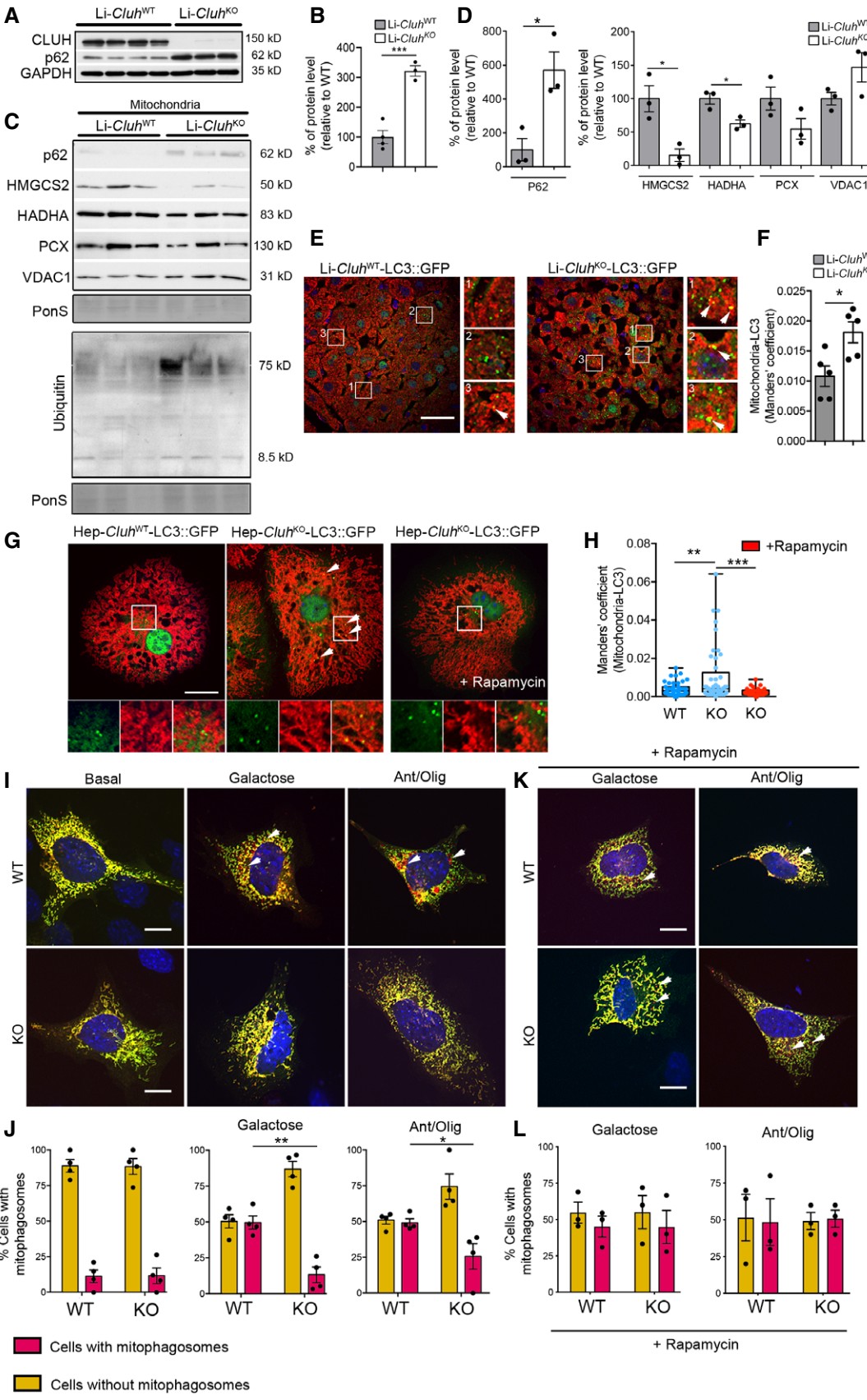

**Figure 8.**

**Figure 8.  CLUH promotes mitophagy.**

A  Western blot analysis of total protein extracts from livers of 8-week-old Li-*Cluh*<sup>WT</sup> and Li-*Cluh*<sup>KO</sup> mice probed with indicated antibodies. GAPDH was used as loading control.

B  Quantification of Western blot shown in (A). WT: $n = 4$; KO: $n = 3$.

C  Western blot analysis of isolated mitochondria from Li-*Cluh*<sup>WT</sup> and Li-*Cluh*<sup>KO</sup> mice probed with indicated antibodies. Ponceau S staining was used as a loading control.

D  Quantification of Western blot shown in (C) ($n = 3$ mice per genotype).

E  Confocal images of liver cryosections derived from Li-*Cluh*<sup>WT</sup> and Li-*Cluh*<sup>KO</sup>-LC3-GFP mice and stained with anti-TOM20 antibody. On the right side, indicated boxes were $3.5\times$ magnified. Arrows point to colocalizing spots. Scale bar, 25 μm.

F  Manders' colocalization coefficient between TOM20 and LC3 from experiments shown in (E). Five images were analyzed per animal to get an averaged value ($n = 5$ mice per genotype).

G  Confocal images of primary hepatocytes derived from Li-*Cluh*<sup>WT</sup> and Li-*Cluh*<sup>KO</sup>-LC3-GFP mice and stained with anti-TOM20 antibody. When indicated, cells were treated with rapamycin (200 nM for 4 h). Bottom panels show $2.3\times$ magnified areas for each channel of indicated boxes. Arrows point to colocalizing spots. Scale bar, 10 μm.

H  Manders' colocalization coefficient from experiment shown in (G) ($n \geq 50$ cells isolated from 3 mice per genotype).

I  Confocal images of WT and *Cluh* KO MEFs transfected with mCherry-GFP-FIS$_{101-152}$ and grown for 16 h in galactose medium or in medium containing 10 μM antimycin A/oligomycin. Arrows indicate mitophagosomes. Scale bar, 10 μm.

J  Quantification of percentage of cells with mitophagosomes. Cells were considered positive when red signal was clearly recognizable ($n = 3$–5 independent experiments, > 100 cells per experiment).

K  Confocal images of WT and *Cluh* KO MEFs transfected with mCherry-GFP-FIS$_{101-152}$, incubated with 200 nM rapamycin, and grown for 16 h in galactose medium or in medium containing 10 μM antimycin A/oligomycin. Arrows indicate mitophagosomes. Scale bar, 10 μm.

L  Quantification of percentage of cells with mitophagosomes. Cells were considered positive when red signal was clearly recognizable ($n = 3$–5 independent experiments, > 100 cells per experiment).

Data information: In (B, D, F, J, L), data are presented as histograms showing the mean ± SEM. In (H), data are presented as boxplots showing the median, the first quartile, and the third quartile. Error bars show minimum and maximum values. (B, D, F) ***$P \leq 0.001$; *$P \leq 0.05$ (Student's *t*-test). (H, J, L) *$P \leq 0.05$; **$P \leq 0.01$; ***$P \leq 0.001$ (one-way ANOVA, Tukey's multiple comparison test).

Source data are available online for this figure.

thousands of mRNAs (so called RNA regulons), by defining distinct compartments for post-transcriptional regulation (Anderson & Kedersha, 2009; Protter & Parker, 2016; Gomes & Shorter, 2019). The CLUH granules observed in hepatocytes are the first example of RNP particles specifically regulating the translation of mRNAs encoding mitochondrial proteins (here shown for *Pcx*, *Hadha*, and *Hmgcs2*). Although they contain G3BP1 and G3BP2, several evidences indicate that the CLUH granules are distinct from SGs: (i) They incorporate puromycin; (ii) they are resistant to CHX treatment; and (iii) they form in the absence of G3BP1 and G3BP2 upon CLUH overexpression.

Intriguingly, CLUH granules are dynamic and display a transcript-specific temporal activation of translation: In fact, *Pcx* and *Hadha* mRNAs are detected in translationally proficient granules mainly upon nutrient-rich conditions, while *Hmgcs2* mRNA upon starvation, in agreement with the prevalence of anaplerotic reactions or the need to synthesize ketone bodies in the respective condition. Thus, cycles of translation can engage different transcripts at different times in the CLUH granules, to match the translational profile to the cellular needs depending on nutrient availability. In addition, we found that a subset of CLUH granules can incorporate puromycin despite inhibition of the transition between translational initiation and elongation by HHT, suggesting that they might contain translationally stalled mRNAs. So far, none of the tested mRNAs could be detected in these granules, opening the question whether they regulate a different group of transcripts. Future studies will be required to fully understand what distinguishes a translationally silent, active, or stalled CLUH granule, and how the transition between these granules is mechanistically regulated.

We show that CLUH granules play crucial signaling roles, and modulate the nutrient-sensing mTORC1 signaling (Saxton & Sabatini, 2017). CLUH granules recruit mTOR and safeguard hepatocytes

against a premature reactivation of mTORC1 during starvation that poses the cells at risk of apoptosis if energy-consuming anabolic pathways are stimulated despite energy stress (Mikeladze-Dvali *et al*, 2005; Teleman *et al*, 2005; Kim *et al*, 2008; Choo *et al*, 2010; Demetriades *et al*, 2014). Importantly, genes involved in transcription (such as *Tfam*) and translation of mtDNA (mitoribosomal proteins) are known to be translationally regulated by mTORC1 via 4E-BPs (Morita *et al*, 2013, 2015), which are consistently hyperphosphorylated in the absence of CLUH. Our data reveal an unexpected dual regulation of hepatocyte mitochondrial metabolism by CLUH during starvation. On the one side, CLUH ensures stability and translation of a group of transcripts involved in mitochondrial catabolic (fatty acid oxidation and amino acid degradation) and ketogenic pathways, while, on the other side, it inhibits the mTORC1-dependent translation of mRNAs involved in mitochondrial protein synthesis, thus reducing energy-consuming anabolic processes. Surprisingly, several mRNAs previously found in CLUH immunoprecipitation experiments (Gao *et al*, 2014) belong to this last category, opening the possibility that CLUH plays a direct role in regulating the translation of these genes. How is this dual function of CLUH regulated, and what is the role of CLUH when nutrients are abundant or in tissues in which mitochondria are not specialized to respond to nutrients remain to be determined. It is interesting that a recent study found that Clu forms granules in *Drosophila* ovaries upon insulin stimulation. However, so far it is not known whether these granules contain mRNAs and are translationally active (Sheard *et al*, 2019).

Besides controlling protein synthesis and anabolic pathways, mTORC1 activation inhibits autophagy. Thus, it was surprising to find that in the absence of CLUH, bulk autophagy was not impaired, but rather enhanced. Induction of bulk RNA degradation by autophagy has been previously observed in hepatocytes upon amino acid starvation (Lardeux *et al*, 1987; Heydrick *et al*, 1991). We speculate

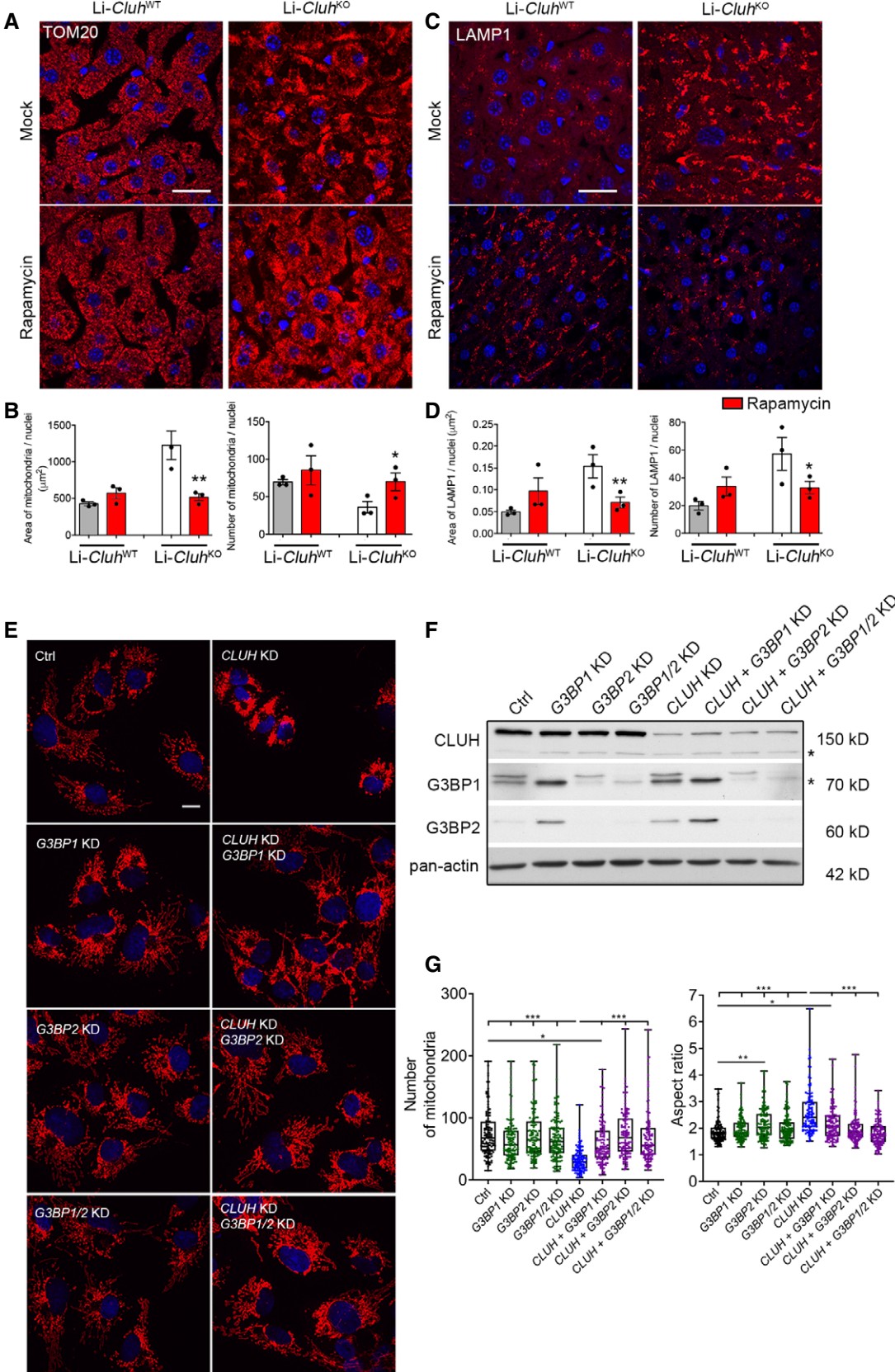

Figure 9.

**Figure 9.   Inhibition of mTORC1 and downregulation of G3BPs recover mitochondrial clustering in the absence of CLUH.**

A   Confocal images of liver cryosections from Li-*Cluh*^WT and Li-*Cluh*^KO mice injected with rapamycin or mock solution and stained with anti-TOM20 antibody. Scale bar, 25 μm.
B   Quantification of morphological parameters of experiments shown in (A). Four images were quantified per animal and averaged. Nuclear staining was excluded for the analysis (*n* = 3 mice per genotype).
C   Confocal images of liver cryosections from Li-*Cluh*^WT and Li-*Cluh*^KO mice injected with rapamycin or mock solution and stained with anti-LAMP1 antibody. Scale bar, 25 μm.
D   Quantification of morphological parameters of experiments shown in (C). Four images were quantified per animal and averaged. Nuclear staining was excluded for the analysis (*n* = 3 mice per genotype).
E   Confocal images of COS-7 cells downregulated for the indicated genes and stained with anti-TOM20 antibody. Scale bar, 10 μm.
F   Representative Western blot of COS-7 cells downregulated for the indicated genes of experiments shown in (E) and probed with indicated antibodies. Asterisks indicate unspecific signal or signal of previous incubation.
G   Quantification of morphological parameters of experiments shown in (E) (*n* > 100 cells from 3 independent experiments).

Data information: In (B, D), data are presented as histograms showing the mean ± SEM. In (G), data are presented as boxplots showing the median, the first quartile, and the third quartile. Error bars show minimum and maximum values. (B, D) **$P \leq 0.01$; *$P \leq 0.05$ (Student's *t*-test). (G) ***$P \leq 0.001$; **$P \leq 0.01$ *$P \leq 0.05$ (one-way ANOVA, Tukey's multiple comparison test).
Source data are available online for this figure.

that sorting of target mRNAs into CLUH granules has a crucial role not only for their translation, but also to protect them against premature decay and autophagic degradation. Consistently, pathways associated with mRNA decay and the lysosome were enriched in proteomics analysis of starved KO hepatocytes. Furthermore, G3BP1-positive granules colocalized with LAMP1-positive organelles upon HBSS starvation with faster dynamics in KO hepatocytes. Whether the decreased occurrence of G3BP1-positive granules in hepatocytes lacking CLUH is totally explained by their faster turnover by autophagy or is also caused by failure of nucleation of the granules is a question to be addressed in the future. Furthermore, it is still unclear whether the role of CLUH in stabilizing target mRNAs is dependent on granule formation.

While inhibiting autophagic degradation of RNP particles containing transcripts for mitochondrial proteins, CLUH positively regulates turnover of mitochondria both *in vivo* in the liver and *in vitro* in hepatocytes and MEFs. Our data are consistent with studies in the fly showing increased levels of p62, mitochondrial recruitment of parkin, increased parkin–Pink1 interaction, and a decreased mitochondrial clearance in the absence of clueless (Sen *et al*, 2015; Wang *et al*, 2016). Importantly, during starvation, mitochondria are protected from macroautophagic degradation for a longer time with respect to cytosolic components or other organelles, for example, ribosomes (Kristensen *et al*, 2008; Rambold *et al*, 2011). However, in the starved liver it may be important to allow a basal level of mitochondrial turnover to replace organelles with a metabolic profile suited for conditions of high nutrients with those specified to perform catabolic functions. Therefore, differential regulation of bulk autophagy and mitophagy may be advantageous in the liver when nutrients are low. In this situation, CLUH plays a central role by coupling mitochondrial biogenesis and mitochondrial turnover to maintain the quality of liver mitochondria and adapt it to cellular needs. A similar concept is currently emerging also from studies implicating PINK1 not only in mitophagy (Lazarou *et al*, 2015), but also in localized translation at the surface of mitochondria (Gehrke *et al*, 2015).

The apparent contrasting finding on autophagy and mitophagy observed in cells and tissues lacking CLUH recapitulates the role of Bre5p, the yeast G3BP orthologue, which is required together with the ubiquitin protease Ubp3 for promoting ribophagy and other forms of autophagy, while inhibiting mitophagy (Ossareh-Nazari

*et al*, 2010; Muller *et al*, 2015). We speculate that by recruiting G3BPs to granules, CLUH restricts their function. Consistently, concomitant depletion of CLUH and G3BPs prevented the formation of mitochondrial clustering upon CLUH downregulation, assigning for the first time to mammalian G3BPs a role in mitophagy regulation. This potentially broadens up the signaling function of CLUH granules to regulation of deubiquitination processes required for specialized forms of autophagy, via G3BPs. G3BP1 and G3BP2 are widely studied for their role in RNA metabolism and SG formation, but they have been associated with multiple functions, including Ras signaling, NFκB activation, and interferon-mediated signaling (Alam & Kennedy, 2019). Unraveling the role of G3BP1 and G3BP2 in mitophagy and regulation of the translation of mRNAs for mitochondrial proteins will be necessary to fully understand the role of CLUH.

Finally, we show that rapamycin administration improved mitochondrial clustering and rescued lysosomal accumulation in *Cluh*-deficient livers, confirming the hypothesis that mitochondrial clustering is a secondary phenotype deriving from impaired turnover of dysfunctional mitochondria and perturbed signaling responses. Interestingly, promoting mitophagy by parkin overexpression also rescues the clustering phenotype in the fly muscle and ovary (Sen *et al*, 2015; Wang *et al*, 2016).

In conclusion, we unveil a complex CLUH-dependent post-transcriptional mechanism of physiological significance in hepatocytes. CLUH coordinately regulates mTORC1 signaling, the catabolic capacity of mitochondria and their turnover to efficiently program metabolism to cope with energy levels (Fig EV5). In the absence of CLUH, hepatocytes are susceptible to cell death upon starvation. In general terms, our data highlight the importance of compartmentalization for post-transcriptional regulation of mitochondrial RNA regulons.

## Materials and Methods

### Animal experiments

The liver-specific *Cluh*-deficient mice (Li-*Cluh*^KO) used in this study were generated in a C57BL/6N congenic background and were described and characterized previously (Schatton *et al*, 2017). When

specified, mice were crossed with LC3-GFP transgenic mice (Mizushima *et al*, 2004) to visualize autophagosomes. All lines were maintained in a pure C57BL/6N background. In all experiments, both male and female mice at 8 weeks of age were used, unless stated otherwise. As controls, *Cluh*^fl/fl^ littermates from the same strain without the *Cre* allele were used (here named WT). Mice were maintained in individually ventilated cages with specified pathogen-free hygiene levels, kept under a 12-h/12-h dark/light cycle, given a regular chow diet *ad libitum* (Sniff V1554-300), and monitored regularly for signs of suffering. Starvation was induced by removing access to food but not water for 24 h. For tissue collections, mice were sacrificed by cervical dislocation. For histological studies, adult mice were anesthetized intraperitoneally with xylazine/ketanest (10 mg/100 mg per kilogram body weight) and perfused intracardially with 4% PFA/PBS.

Before the first injection of rapamycin or mock solution, mice were starved for synchronization. Animals were allocated randomly to treatments. Rapamycin was dissolved in DMSO to 100 mg/ml and then further in 5% PEG-1500 and Tween-20 in water until 1.2 mg/ml. Solution was filtered through a 0.2-μm filter. Rapamycin (7.5 mg/kg) or mock (5% PEG-1500, 5% Tween-20) was injected daily for 1 week. After this time, animals were sacrificed by cervical dislocation and tissues were collected for analysis. For immunostaining, tissues were kept in 4% PFA/PBS for 5 days before cutting in cryostat. All animal procedures were performed in accordance with European Union (EU directive 86/609/EEC), national (Tierschutzgesetz), and institutional guidelines and were approved by local authorities (Landesamt für Natur, Umwelt, und Verbraucherschutz Nordrhein-Westfalen, Germany).

### Hepatocyte isolation

To isolate primary hepatocytes, adult mice at 8 weeks of age were anesthetized as specified above and perfused through the vena cava with EBSS w/o $Ca^{2+}$ and $Mg^{2+}$ supplemented with 0.5 mM EGTA, followed by EBSS including $Ca^{2+}$ and $Mg^{2+}$ containing 10 mM HEPES, 0.4 mg/ml collagenase type 2 (Worthington), and 0.04 mg/ml trypsin inhibitor (Sigma-Aldrich) at 37°C. Livers were dissected, and single-cell suspensions were generated through a 70-μm nylon filter. Cells were then mixed with Percoll/HBSS solution [9 vol Percoll (GE Healthcare) + 1 vol 10× HBSS buffer] and centrifuged at 1,000 × *g* for 7 min. Pellets were washed twice with 5% FBS/DMEM and plated on coverslips coated with fibronectin (Thermo Fisher Scientific). Primary hepatocytes were maintained in 4.5 g/l glucose DMEM including 5% FBS, 2 mM L-glutamine, and 2% penicillin/streptomycin (basal medium). Experiments in hepatocytes were performed 24 h after plating.

### RNA *in situ* hybridization

Hepatocytes were fixed with 10% neutral buffered formalin for 15–30 min. Fixed cells were dehydrated in 5-min series of 50-70-100% ethanol and stored at −20°C until use. Single-molecule RNA *in situ* hybridization was performed using the RNAscope 2.5 HD Fluorescent Reagent Kit (Advanced Cell Diagnostics, Inc.). Coverslips were rehydrated in 5-min series 100-70-50% ethanol and a final 15-min incubation in PBST (PBS and 1% Tween-20), and treated with Protease III for 30 min at room temperature before

hybridization with the corresponding probe. Target probes to detect murine *Hadha* (#459331), *Pcx* (#418831), *Hmgcs2* (#437141), and *Actb* (#316741) were designed by the manufacturer. *Pcx*, *Hadha,* and *Hmgcs2* were selected in C1 channel while *Actb* in C2 channel. Signal was amplified according to the manufacturer's instructions and detected with an Amp4 Alt A-FL probe, which gives fluorescence in 488 nm for the probes in C1 or Amp4 Alt B-FL probe which gives fluorescence in 488 nm for probes in C2 (Advanced Cell Diagnostics, Inc.). When combined with immunostaining, coverslips were washed with PBST followed by blocking and incubation with primary antibodies. Nuclei were stained with DAPI. Samples were analyzed using a spinning-disk confocal microscope (UltraVIEW VoX, PerkinElmer) with a Plan Apochromat total internal reflection fluorescence 60 Å~/1.49 oil DIC objective. Colocalization analyses were carried out using JACoP plugin (Bolte & Cordelieres, 2006). Manders' coefficient was used to quantify colocalization (Manders *et al*, 1993).

### Ribopuromycylation assay

Hepatocytes were incubated for 5 min at 37°C with 91 μM puromycin (Sigma-Aldrich) and 208 μM emetine (Sigma-Aldrich). Dishes were transferred to ice and washed twice with ice-cold PBS containing 355 μM CHX. Plates were incubated for 2 min on ice with ice-cold PB Buffer [50 mM Tris–HCl pH 7.5, 5 mM $MgCl_2$, 25 mM KCl, 355 μM cycloheximide, 10 U/ml RNase OUT (Life Technologies), 0.007% digitonin with protease inhibitor EDTA-free (Roche)] and washed twice with ice-cold WB Buffer [50 mM Tris–HCl pH 7.5, 5 mM $MgCl_2$, 25 mM KCl, 355 μM cycloheximide, 10 U/ml RNase OUT (Life Technologies), 20 mM sucrose with protease inhibitor EDTA-free (Roche)]. Hepatocytes were fixed with 10% formalin for 15 min. Immunofluorescence was performed as described previously, using an anti-puromycin antibody (#MABE343, clone 12D10, EMD Millipore) to detect active translation sites. When combined with RNA *in situ* hybridization, RNAscope protocol was performed prior to immunofluorescence. Samples incubated without puromycin were used as a control for antibody specificity. To inhibit translation, cells were treated with 500 μM arsenite for 30 min or with 2 μg/ml HHT for 15 min before ribopuromycylation was performed. Fluorescence intensity was visualized with mpl-inferno lut from ImageJ. Colocalization analyses were carried out using JACoP plugin (Bolte & Cordelieres, 2006). Manders' coefficient was used to quantify colocalization (Manders *et al*, 1993). To quantify puromycin particles in HBSS, ImageJ was used first to remove nuclear signal from the images. Later, the puromycin channel was manually thresholded and the watershed plugin was used to separate particles. Puromycin particles bigger than 5 pixels were analyzed, using the "analyze particle" feature of ImageJ. To detect puromycin granules also positive for CLUH, the CLUH channel was first thresholded and the puromycin granules positive for CLUH were counted. The intensity profiles shown in Fig 3 were performed in the original images drawing a line of 100 pixel long and 10 pixels thick.

### Immunofluorescence analysis

Cells were fixed in 4% PFA/PBS for 15 min and stored at 4°C in PBS. HeLa and COS-7 cells were permeabilized with 0.2% Triton

X-100 in PBS for 20 min, blocked for 30 min (5% milk, 2.5% BSA, 10% serum, 0.2% Triton X-100 in PBS), and incubated with the corresponding primary antibody overnight at 4°C or 2 h at room temperature. Antibodies used for immunofluorescence were poly-clonal rabbit anti-CLUH (#ab178342, Abcam; # ARP70642_P050, Aviva), polyclonal rabbit anti-LAMP1 (#ab24170, Abcam), poly-clonal rabbit anti-TOMM20 (#sc-11415, Santa Cruz Biotechnology), monoclonal mouse anti-G3BP1 (#sc-81940, clone TT-Y, Santa Cruz Biotechnology), monoclonal mouse anti-TIA-1 (#sc-166247, clone G3, Santa Cruz Biotechnology), monoclonal mouse anti-FLAG (#F3165, clone M2, Sigma-Aldrich), monoclonal mouse anti-puro-mycin (#MABE343, clone 12D10, EMD Millipore), monoclonal rabbit anti-mTOR (#2983, clone 7C10, Cell signaling), monoclonal mouse anti-DCP1A (#H00055802-M06, clone 3G4, Abnova), poly-clonal rabbit anti-G3BP2 (#ab86135, Abcam), and polyclonal rabbit anti-CLUH (#NB100-93306, Novus Biologicals). Secondary antibod-ies were diluted in blocking buffer and incubated for 1 h in the dark at room temperature. Coverslips were washed with PBS and mounted with Fluorsave (EMD Millipore) and visualized with spin-ning-disk confocal microscope (UltraVIEW VoX, PerkinElmer) using a 60× objective. Microscopy chromatic aberration was tested regu-larly for spinning-disk microscope using standard samples with 100- and 500-nm beads. Images acquired with confocal microscope were deconvoluted using ImageJ (NIH) prior to analysis to increase reso-lution. Brightness was adjusted equally in the entire images. The amount, the relative size, and area of LC3 and LAMP1 particles were analyzed using a plugin developed by Dagda et al (2008).

### Tissue histology, immunohistochemistry, and immunofluorescence

Livers of perfused mice were post-fixed for 2 days in 4% PFA/PBS at 4°C. For cryostat sectioning, after fixation tissues were dehydrated in 15% sucrose for 6 h and 30% sucrose overnight. After this treat-ment, liver pieces were embedded in optimum cutting temperature mounting medium (Tissue-Tek) and stored at −80°C. Before cutting, samples were equilibrated to −20°C for 4 h and cut in 10-μm sections. Immunofluorescence was performed following a standard protocol. Sections were washed in PBS, blocked in 1% Western Blocking Reagent (Roche), and then incubated with indicated anti-bodies in blocking reagent overnight at 4°C. Afterward, sections were washed and incubated in secondary antibodies diluted in block-ing buffer for 1 h at RT, washed again, stained with DAPI (Sigma-Aldrich), and mounted with Fluorsave (EMD Millipore). Antibodies used in tissue immunofluorescence were polyclonal rabbit anti-CLUH (# ARP70642_P050, Aviva), polyclonal rabbit anti-LAMP1 (#ab24170, Abcam), and polyclonal rabbit anti-TOMM20 (#sc-11415, Santa Cruz Biotechnology). For imaging, a spinning-disk confocal microscope (UltraVIEW VoX; PerkinElmer) with a Plan Apochromat total internal reflection fluorescence 60 Å~/1.49 NA oil DIC objective was used.

Analysis of apoptosis in liver was performed in paraffin-embedded samples. Paraffin liver sections were rehydrated, and antigen retrieval was performed by heating in 10 mM sodium citrate and 0.05% Tween 20, pH 6.2. Rabbit polyclonal anti-cleaved caspase 3 (Asp175) (#9661, Cell Signaling) was used to identify apoptotic cells. Biotinylated secondary antibodies were purchased from Vector Laboratories. Stainings were visualized with ABC kit

Vectastain Elite (Vector Laboratories) and DAB substrate (Dako). The number of apoptotic cells was counted and shown relative to tissue area. Image analysis in tissues was performed on 4–5 animals. Four pictures from the same animal were taken, and the values were averaged.

### Cell lines

Cell lines (COS-7, HeLa, and MEF) were cultured in 4.5 g/l glucose DMEM supplemented with 10% FetalClone III serum (Hyclone), 2% penicillin/streptomycin, and 2 mM L-glutamine (Invitrogen), and regu-larly checked for mycoplasma contamination. Cluh-deficient MEFs were previously described (Gao et al, 2014). HeLa cells lacking CLUH were generated by CRISPR-Cas9 technology (Wakim et al, 2017).

### RNA interference and mitochondrial clustering

RNA interference of CLUH in COS-7 cells was done as described previously (Gao et al, 2014). To downregulate G3BP1 and G3BP2, HeLa and COS-7 cells were transfected with 100 nM siRNA (Ctrl siRNA: #D-001206-14-20; G3BP1 siRNA: #M-012099-02-0020; G3BP2 siRNA: #M-015329-00-0020 from Dharmacon) using Lipofectamine 2000 (Life Technologies) on two consecutive days and experiments were performed 72 h after first transfection. Mitochondrial cluster-ing recovery experiments were performed with 200 nM rapamycin or 200 nM torin for 6 h prior fixation. Mitochondrial morphology parameters were obtained using a previously described plugin (Dagda et al, 2009).

### CLUH granules and SG induction

Canonical stress granules were induced in hepatocytes and HeLa cells by incubating with 500 μM arsenite for 30 min. To dissolve stress granules, 0.1 mg/ml cycloheximide was added to the cells for 30 min. To induce CLUH granules in HeLa cells, cells were trans-fected with constructs encoding full-length human untagged CLUH or CLUH-Flag for 48 h with Lipofectamine 2000 (Life Technologies). Full-length human CLUH was cloned into p3xFLAG-CMV-14 or into pcDNA3.1 using HindIII/EcoRI restriction sites. CLUH granules in hepatocytes were induced by HBSS treatment as stated.

### Live imaging expression of G3BP1-GFP

For live imaging, 200,000 HeLa cells were seeded in glass bottom microwell dishes (#P356-1.5-14-C, Mat Tek Corporation) and trans-fected with 0.5 μg G3BP1-GFP plasmid (kindly provided by Dr. Jamal Tazi), using Lipofectamine 2000. Twenty-four h later, cells were incubated in HBSS or treated with arsenite (with or without CHX) and videos were recorded taking 1 picture/min using a spin-ning-disk confocal microscope (UltraVIEW VoX, PerkinElmer) at 37°C and 5% $CO_2$. Only cells showing green fluorescence but not granules produced by G3BP1 overexpression were selected for recording.

### Autophagy and mitophagy experiments

To assess autophagic flux, hepatocytes were treated with or without 100 nM bafilomycin A for 2 h in basal or HBSS medium. To assess

mitophagy, MEFs were transfected with mCherry-GFP-FIS$_{101-152}$ plasmid (Allen *et al*, 2013). After 24 h, mitophagy was induced either by replacing the medium with glucose-free DMEM (#11966-025, Invitrogen) supplemented with 10 mM galactose, 10% dialyzed serum (Invitrogen), 2% penicillin/streptomycin, 2 mM L-glutamine, and 1 mM sodium pyruvate, or incubating the cells with 10 μM antimycin A and 10 μM oligomycin for 16 h. When indicated, 200 nM rapamycin was added. Cells with mitophagosomes were counted when red signal was clearly recognizable (> 100 cells per replicate in 3–5 independent experiments).

### Western blots and mitochondrial purification

Cell pellets were lysed in RIPA buffer [50 mM Tris–HCl pH 7.4, 150 mM NaCl, 1 mM EDTA pH 8.0, 0.25% deoxycholic acid, 1% Triton X-100, and protease inhibitor cocktail (Sigma-Aldrich)], while tissues were lysed in RIPA buffer including 0.1% SDS. Proteins were quantified with standard Bradford (Bio-Rad) assay prior to SDS–PAGE and blotting on PVDF membranes. To isolate mitochondria, livers were dissected and homogenized in Mitochondria Isolation Buffer (MIB, 100 mM sucrose, 50 mM KCl, 1 mM EDTA, 20 mM TES, 0.2% free fatty acid–BSA) at 1,200 rpm until the solution was homogenous. The solution was centrifuged at 8,500 × *g* at 4°C for 5 min. Pellets were washed twice with MIB and centrifuged at 800 × *g*. Clean supernatants were then centrifuged at 8,200 × *g* to obtain pure mitochondria. Final pellets were resuspended in 500 μl MIB w/o BSA, and 10 μg of pure mitochondria were pelleted and resuspended in MIB w/o BSA. The following antibodies were used for Western blot: polyclonal rabbit anti-4E-BP1-pT37/46 (#2855, clone 236B4), polyclonal rabbit anti-4E-BP1(#9452), monoclonal rabbit anti-VDAC1 (#4661, clone D73D12), and polyclonal rabbit anti-RPS6-pS235/236 (#2211) from Cell Signaling; monoclonal mouse anti-GAPDH (#MAB374, clone 6C5) and monoclonal mouse anti-pan-actin (#MAB1501R, clone C4) from EMD Millipore; polyclonal rabbit anti-CLUH (for detection of human CLUH, #NB100-93306), polyclonal rabbit anti-RPS6 (#NB100-1595), and polyclonal rabbit anti-LC3 (#NB100-2220) from Novus Biologicals; polyclonal rabbit anti-CLUH (for detection of murine CLUH, # ARP70642_P050) from Aviva; monoclonal mouse anti-p62 (#H00008878-M01, clone 2C11) from Abnova; monoclonal mouse anti-ubiquitin (#sc-8017, clone P4D1) and monoclonal mouse anti-HMGCS2 (#sc-376092, clone G-11) from Santa Cruz Biotechnology; and polyclonal rabbit anti-HADHA (#ab54477), monoclonal rabbit anti-PCX (#ab128952, clone EPR7365), monoclonal rabbit anti-G3BP1 (#ab181150, clone EPR13986(B)), polyclonal rabbit anti-G3BP2 (#ab86135), and polyclonal rabbit anti-LAMP1 (#ab24170) from Abcam.

### RNA isolation, cDNA synthesis, and quantitative real-time PCR

RNA isolation was performed with TRIzol reagent (Invitrogen) according to the manual instructions. 2 μg total RNA was retro-transcribed with the SuperScript First-Strand Synthesis System (Life Technologies) according to the instructions of the manual. Quantitative real-time PCR was performed with SYBR Green Master Mix (Applied Biosystems) using Quant Studio 12K Flex Real-Time PCR System thermocycler (Applied Biosystems). Each reaction was performed in duplicate. The following primers were used for amplification: *G3bp1* forward: 5′-CCCGACGATGTGCAGAAGAG-3′; *G3bp1* reverse: 5′-CCCAAGAAAACGTCCTCAAGTC-3′; *Rpl13* forward: 5′-AGCCCCACTTCCACAAGGA-3′; *Rpl13* reverse: 5′-TTGCGCCTGCGGATCT-3′; *Psat1* forward: 5′-AGTGGAGCGCCAGAATAGAA-3′; *Psat1* reverse: 5′-CTTCGGTTGTGACAGCGTTA-3′; *Phgdh* forward: 5′-GACCCCATCATCTCTCCTGA-3′; *Phgdh* reverse: 5′-GCACACCTTTCTTGCACTGA-3′; *Mthfd2* forward: 5′-CTGAAGTGGGAATCAACAGTGAG-3′; *Mthfd2* reverse: 5′-GTCAGGAGAAACGGCATTGC-3′; *Hprt* forward: 5′-TCCTCCTCAGACCGCTTTT-3′; and *Hprt* reverse: 5′-CATAACCTGGTTCATCGC-3′. *Rpl13* was used for normalization in hepatocytes and *Hprt* in tissues. Fold enrichment was calculated with the formula: $2^{(-\Delta\Delta Ct)}$.

### RNA-seq

Libraries were prepared using the Illumina® TruSeq® mRNA stranded sample preparation kit. Library preparation started with 1 μg total RNA. After poly-A selection (using poly-T oligo-attached magnetic beads), mRNA was purified and fragmented using divalent cations under elevated temperature. The RNA fragments underwent reverse transcription using random primers, followed by second-strand cDNA synthesis with DNA Polymerase I and RNase H. After end repair and A-tailing, indexing adapters were ligated. The products were then purified and amplified (14 PCR cycles) to create the final cDNA libraries. After library validation and quantification (Agilent 2100 Bioanalyzer), equimolar amounts of library were pooled. The pool was quantified by using the Peqlab KAPA Library Quantification Kit and the Applied Biosystems 7900HT Sequence Detection System. The pool was sequenced by using the Illumina novaSeq6000 instrument and a PE100 protocol. Data were analyzed using QuickNGS (Wagle *et al*, 2015). Reads were mapped to the human reference assembly, version GRCh38, using TopHat2 (Kim *et al*, 2013), and gene quantification was carried out using a combination of Cufflinks (Trapnell *et al*, 2010) and the DEseq2 package (Anders & Huber, 2010) with genomic annotation from the Ensembl database, version 80. The gene lists were filtered according to FC and *P*-value in comparison of the library-size normalized read counts between samples and controls. We used the DESeq2 package from the Bioconductor project to determine means as well as FC and *P*-values. Significance was determined using Wald's test. In contrast, gene expression for the individual samples was calculated by the Cufflinks package and returned as FPKM (fragments per kilobase of transcript per million mapped reads) values.

### LC-MS/MS analysis and bioinformatics

Proteins were extracted in 4% SDS followed by acetone precipitation overnight at −20°C. After resuspension in 6 M urea/2 M thiourea, proteins were reduced with DTT and carbamidomethylated with IAA. LysC (Wako) and trypsin (Promega) were added for overnight digestion. Samples were then desalted using SDB-RPS StageTips as previously described (Rappsilber *et al*, 2003). Proteomic analysis was performed using an Easy nLC 1000 UHPLC coupled to a QExactive Plus mass spectrometer (Thermo Fisher). Peptides were resuspended in Solvent A (0.1% FA), picked up with an autosampler, and loaded onto in-house made 50 cm fused silica columns (internal diameter (I.D.) 75 μm, C18 1.7 μm, Dr. Maisch beads) at a flow rate of 0.75 μl/min. A 240-min segmented gradient of 5–34% Solvent B (80% ACN in 0.1% FA) over 215 min, 34–55%

Solvent B over 5 min, and 55–90% Solvent B over 5 min at a flow rate of 250 nl/min was used to elute peptides. Eluted peptides were sprayed into the heated transfer capillary of the mass spectrometer using a nano-electrospray ion source (Thermo Fisher Scientific). The mass spectrometer was operated in a data-dependent mode, where the Orbitrap acquired full MS scans (300–1,750 $m/z$) at a resolution ($R$) of 70,000 with an automated gain control (AGC) target of $3 \times 10^6$ ions collected within 20 ms. The dynamic exclusion time was set to 20 s. From the full MS scan, the 10 most intense peaks ($z \geq 2$) were fragmented in the high-energy collision-induced dissociation (HCD) cell. The HCD normalized collision energy was set to 25%. MS/MS scans with an ion target of $5 \times 10^5$ ions were acquired with $R = 17,500$, with a maximal injection time of 60 ms and an isolation width of 2.1 $m/z$.

The raw files were processed using MaxQuant software and its implemented Andromeda search engine and LFQ algorithm (Cox *et al*, 2011). Parameters were set to default values. GO annotations, statistical analysis, and Welch *t*-test were performed using Perseus software (Tyanova *et al*, 2016). Significant protein fold changes were determined by permutation-based FDR approach (cutoff 0.05, number of permutations = 500, fudge factor $[S_0] = 0.1$) (Tusher *et al*, 2001). One-dimensional enrichment of GO terms based on $\log_2$ fold changes was performed using the algorithm implemented in Perseus with a Benjamini–Hochberg FDR threshold of 0.02, and resulting scores were used for the 2D score plots. Plots were generated using Instant Clue (Nolte *et al*, 2018).

### Statistical analysis

Data plotted as histograms correspond to the average values and standard error of the means of different animals or of independent experiments. Data from individual cells are presented as boxplot, showing the median, the first quartile, and the third quartile. To compare two groups, two-tailed paired or unpaired Student's *t*-test was performed. When more than two groups were analyzed, one-way ANOVA was performed using Tukey's *post hoc* test. To test statistical difference between genotypes at different time points, two-way ANOVA was performed. *P*-values were calculated using GraphPad Prism software (Version 6.02). Sample size was chosen according to our previous experience. No statistical test was used to predetermine sample size. At least three independent biological replicates were used unless specified otherwise. Values were considered statistically different when $P < 0.05$.

## Data availability

The dataset produced in this study are available in the following databases: Proteomics: Data have been deposited to the ProteomeXchange Consortium via the PRIDE (Perez-Riverol *et al*, 2019) partner repository with the dataset identifier PXD014098 (http://www.ebi.ac.uk/pride/archive/projects/PXD014098).

**Expanded View** for this article is available online.

## Acknowledgements
We are grateful to Esther Barth, and the CECAD imaging and proteomics facilities for excellent technical assistance. We thank Matteo Veronese for helping with data quantification and Karolina Szczepanowska for experimental advice. We thank Thomas Langer for critical reading of the manuscript and fruitful discussion. This work was supported by grants from the Deutsche Forschungsgemeinschaft (RU 1653/2-1 and SFB1218/A05) to E.I.R.

## Author contributions
Funding Acquisition, EIR; Conceptualization, DP-M, DS, and EIR; Investigation and Formal Analysis, DP-M, DS, M-CM, and JLW; Resources, SK; Analysis of MS Data, JLW; Visualization, DP-M, DS, M-CM, and JLW; Writing—Original Draft, DP-M, DS, and EIR; Writing—Review & Editing, all authors; and Supervision, MK and EIR.

## Conflict of interest
The authors declare that they have no conflict of interest.

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
