## [Review Process File · The EMBO Journal]

CLUH granules coordinate translation of mitochondrial proteins with mTORC1 signaling and mitophagy

David Pla-Martín, Desiree Schatton, Janica Wiederstein, Marie-Charlotte Marx, Salim KHIATI, Marcus Krüger, and Elena Rugarli.

Review timeline:

Submission date:	20 th June 2019
Editorial Decision:	31 st July 2019
Revision received:	27 th December 2019
Editorial Decision:	31 st January 2020
Revision received:	5 th February 2020
Accepted:	7 th February 2020

Editor: Elisabetta Argenzio

Transaction Report:

1st Editorial Decision

31st July 2019

Thank you for submitting your manuscript entitled "CLUH-dependent RNA granules coordinate mitochondrial catabolic pathways with mTORC1 signaling and mitophagy" to The EMBO Journal. Please accept my apologies for the unusual length of the review process, due to the delayed arrival of some reports and detailed discussions within the team. Your study has been sent to three referees for evaluation, whose reports are enclosed below.

As you can see, the reviewers concur with us on the overall interest of your findings. However, they also raise several key points that need to be addressed before they can support publication in The EMBO Journal. We find that the referees' requests are both fair and important, and require you to add new experimental data, appropriate quantifications and statistical analyses, in addition to editing the text. While recognizing that a major revision is needed to solve the referee points, we remain interested in your study and would thus like to invite you to revise the manuscript in response to the referees' reports.

I should also note that conclusively addressing all major and minor issues raised by the referees would be essential for publication in The EMBO Journal, as well as a strong support from the reviewers.

REFeree REPORTS

Referee #1:

OVERVIEW

This is an interesting and novel study, building upon the authors' earlier work on CLUN, a protein

which forms 'granules' that contain certain mRNAs. The authors extend their earlier work to study roles for CLUH in modulating mitochondrial metabolism, mitophagy (whereby defective mitochondria are removed) and the mTORC1 pathway (which negatively regulates mitophagy/autophagy).

Their data raise a number of issues which the authors are asked to address; the length of this list reflects the diverse nature of the kinds of data shown, and the fact that this is indeed an interesting investigation:

MAJOR POINTS

1. In their colocalization studies (Fig. 1E,F; Fig. 2B), the authors make use of Mander's co-efficients; the spread of coefficients in the figures is wide - can the authors provide statistical analysis to assess whether the changes they see are actually statistically significant?
2. It seems a bit confusing that Pcx and Hadha associate with granules more upon starvation, but in the presence of glucose the granules do exist and the mRNAs seem to be translated. Are these mRNAs truly translated when associated with these granules (which would contrast with the situation for P-bodies and stress granules, in which mRNAs are not actively translated)? I.e., do puromycin data speak to the actual synthesis of these proteins within the granules or to other components of the complexes? Given that puromycin labeling is used, these cannot presumably be complete proteins, so that should rule out that they have been made elsewhere and subsequently joined the CLUH granules.
3. Bottom of p. 9 and Fig. 3B - "starvation-induced granules" - the authors same the number of granules +/- starvation, so the granules are not induced by starvation. They can only conclude that the granules containing G3BP1 mRNA are partly dependent on CLUH (some may be other kinds of granules, I guess).
4. Bottom of p. 9 and Fig. S3A: the authors cannot conclude that CLUH induces granules without data for control cells (to see if there are MORE granules in the cells that overexpress CLUH. It is not really clear what Fig. S3B shows - only really shows that ectopically overexpressed FLAG-CLUH associates with granules, which is not surprising, as endogenous CLUH does this).
5. P. 10, the opening statement is very far-reaching - what is the justification for this? ('organize the cellular metabolic response to nutrient availability').
6. Related to Fig. 3: the use of the terms 'upregulated' and 'downregulated' is misleading as it implies an active process. Presumably, loss of CLUH causes destabilisation of some mRNAs (cf. their earlier paper, Schatton et al., ref. 18), for example, in which case the authors should just say 'was decreased'.
7. Fig. S4B - some of the changes they report are small, e.g., $\text{antilog}_2 0.25$, which is <1.2 fold change - what can they conclude about the biological significance of such small significance; validation would help.
8. Fig. 3G - needs better information on which mitochondrial enzymes changed, and the average extent. Fig. 3F does not show what happens to mitochondrial R-proteins in general (only four are pinpointed), although they are mentioned in the text; why? Also need to identify or give examples of 'mitochondrial enzymes involved in catabolic pathways' (just giving gene names is inadequate).
9. Page 13 and Fig. 4: they don't mean they saw 'an increase in phosphorylation of 4E-BP1 and PRAS40 upon 24 h starvation'; their phosphorylation actually falls, but less so in the CLUH-KO cells. This part is confusingly written - there is no hyperactivation of mTORC1, it is just less sensitive to inhibition by nutrient starvation.
10. In Fig. 3G,H - why do the authors use Cos-7 siRNA cells, rather than cells from the KO mice?
11. Fig. 4A and B: The authors need to show G3BP1-mTOR/RAPTOR co-localization under basal conditions. Also, for these and other instances of imaging figures, it'd be helpful - and much more

convincing - to show a population of cells (more than 5) rather than just showing one single cell.

12. On page 13, the authors claim that "Furthermore, in basal conditions the granules were next to LAMP1-positive structures, where mTORC1 is active, while they colocalized with LAMP1 in HBSS (Fig. 4B)." However, as far as I can see from Fig. 4B, G3BP1 is pretty much dispersed everywhere within the cell and not just localized with LAMP1. The authors really need to show some enlarged areas (like in Fig. 2A) and present the data with stats to support their claim (if true) here.

13. In Fig. 4C and E, the authors need to show the phosphorylation status of some other conventional mTORC1 downstream targets such as S6K1 or rpS6, like they did in G and H. I can understand P-S6K1 is difficult to look at in primary cells, but at least P-rpS6 should be an easier one; also very important to quantify P-protein/total protein for each one across at least three experiments (e.g., for 4E-BP1). The same applies to Fig. 4G/H. This is crucial as the claimed effects are modest/subtle. The blot for 4E-BP1 pT37/46 is of poor quality (Minor point: actually the residue numbers are T36/T45 in mice). In the title for legends of Fig. 4, "regulate" sounds quite vague, better change to a more precise word like "contributes to" or "prevents".

14. Regarding the co-localisation (or adjacency) of CLUH with LAMP1 - given that the authors show effect on signaling downstream of mTORC1, which is activated on the lysosomal surface: is the lysosomal association of mTORC1 altered in CLUH-KO vs. WT cells?

15. Fig. S5E - what are the different bars in the graphs? This is not explained in the legend.

16. The observation that rapamycin restores mitophagy in CLUH-deficient cells does not, as claimed, prove that the effect of CLUH is via mTORC1 - it just shows that the effect of lack of CLUH is overcome by rapamycin, a drug known to promote autophagy through mTORC1 inhibition.

17. Fig. 5K: "Instead, mitochondrial proteins encoded by CLUH target mRNAs were decreased in purified mitochondria (Fig. 5K)." This can be visualised in the COX IV blot, although this is less convincing with respect to NDUF9, and not at all with ATP5a. Therefore, I am not sure the general statement the authors make is a valid one.

18. "In further agreement with a mitophagy block, LC3 signal co-localized more with mitochondria in liver tissue lacking CLUH compared to controls (Fig. 5L, M)." The graph of one of the Manders' coefficients in 5M definitely indicates this, but the two images in 5L do not look much different from each other.

19. "An accumulation of bigger autophagosomes was confirmed in primary hepatocytes derived from Li-CluhKO mice (Fig. 6A-C). Furthermore, also in cells more LC3-positive structures co-localized with mitochondria, and rapamycin treatment was effective in restoring the co-localization between mitochondria and LC3 puncta to WT levels (Fig. 6D, E)." Again, with nonetheless much variation, the graphs (6B, C and E) indicate this, but it is difficult to discern much difference between each of the three pictures in 6D (in particular WT v.s KO vs. KO + rapamycin, representative of 6E).

20. One item that would strengthen the 'story' is the classic without and with CQ/Bafilomycin assay, with/without CLUH expression, followed by immunoblotting for p62 and LC3. This could be inserted into Figure 5. This would confirm that there is a block in autophagic flux, since several of the other results could arise from increased synthesis rather than a block in degradation.

MINOR POINTS:

21. What does CLUH-starved hepatocytes mean (p. 10, last line)?

22. Minor point: p. 15 - torin 1 is an mTOR kinase inhibitor rather than an irreversible inhibitor. It works differently from rapamycin.

Referee #2:

The article of Pla-Martin et al joins the series of beautiful papers from Rugarli-lab in identifying and characterising CLUH as an RNA-binding protein for nuclear-encoded mRNAs of proteins to be translocated to mitochondria, and as a major regulator of metabolism, tightly communicating with mTOR complex to fine-tune mitochondrial and cytoplasmic translation in variable nutrient situations.

The authors have previously shown that CLUH binds a subset of nuclear-encoded transcripts and promotes their stability and translational output involved in OXPHOS, TCA cycle, amino acid degradation as well as ketogenesis. Here they show that the membraneless organelle-like G3BP1-positive RNA-granule-structures function in close vicinity to mTORC1 complex, and that CLUH is the driver of these RNA-granules. They propose that CLUH has a complex interplay with mTORC1, suppressing mTORC1 from overactivity at the time of early starvation. Utilizing hepatocyte cultures and mouse liverKO models, the authors report that nutrient state affects mRNA binding to the CLUH granules and that the granules are not only for mRNA storage, but translationally active. Furthermore, their RNAseq experiments of KO and WT cells in starved and fed conditions suggest that CLUH promotes mitochondrial catabolic functions, but inhibits mitochondrial translation.

The paper is well-written and the results are of high quality and provide novel information of coordination of translation in starved and fed state. The supplementary Tables did not exist in the version that I got for review.

Comments:

The proteomics-data suggest that under starved conditions, CLUH-KO cells upregulate mitochondrial translation -associated proteins. As this does not yet indicate activation of translation, which by itself would be somewhat surprising, I would like to see a mitochondrial translation assay, to test whether the translational activity is induced. The potential function of CLUH as a cytoplasmic regulator of mitochondrial ribosomal function would be very interesting.

The experiments of auto/mitophagy do not allow the conclusions made of CLUH participating in regulator of mitophagy regulation together with mTORC1. I do agree that that CLUH-KOs seem to have stalled autophagic flux (p62 accumulation, LC3 puncta) and this is resolved by rapamycin. However, this result is not necessarily an indication of a CLUH-mTORC1 functional link, because the lack of CLUH causes deficiency of oxphos enzymes (as visible in fig 4K, for oxphos complexes CI, IV and V), presumably because of lack of nuclear-encoded oxphos subunits. mTORC1 has been shown to activate upon oxphos deficiency, and rapamycin ameliorates those phenotypes (e.g. Kaeberlein lab) probably by allowing auto/mitophagy. Therefore, roles of CLUH for mitophagy are likely to be secondary. Induction of autophagy may also explain the improved viability that they see in rapamycin treated CLUH-KOs. The authors should revise the text to be less definitive on the mechanistic link of CLUH in mitophagy/autophagy caused by mitochondrial dysfunction.

For apoptosis induction it would be good to see Western blots of starved mouse livers and cell lines, to get an overall view of the extent of caspase-3 cleavage.

Minor comments: the abstract has a lot of abbreviations not clear for experts of this topic. They should be explained and written open.

Referee #3:

In this paper, the authors investigate the mechanisms by which the RNA Binding protein CLUH (clustered mitochondria homolog) in modulating the expression of mitochondrial genes and its role during nutrient fluctuation. In response to nutrient, starvation mitochondria change its gene expression profile to adapt to this stressful condition and ensure energy production and cell survival. In previous work, that authors' lab demonstrated that CLUH regulates the expression of transcripts encoding key factors that affect mitochondria function and adaptation to starvation. In this work, the authors provide a series of observations suggesting that in response to nutrient starvation, CLUH

form specific cytoplasmic RNA granules, via which it modulates mitochondria gene expression and fate. They suggest that the CLUH-granules are different from Stress Granules, even though they recruit G3BP1 and TIA-1, two bonafide SG promoters. In the situation where nutrients are available and abundant, mTORC1 plays a key role in promoting the expression of factors involved in oxidative phosphorylation and activation of mitochondrial translation, the expression of many of which is regulated by CLUH. The authors provide experiments showing that CLUH, possibly via its granules, interferes with the premature activation of the mTORC1 pathway in response to nutrient starvation, leading to the inhibition of mitochondria anabolic pathways. Indeed, depleting CLUH, which in turn prevents the formation of CLUH-granules, triggers the hyperactivation of mTORC1 leading to cell death.

In their JCB 2017 paper, the authors demonstrated the importance of CLUH proteins in regulating mitochondrial function and adaptive capacity to help the cell deal with stressful conditions such as nutrient starvation. Hence, delineating the mechanisms behind the CLUH-mediated effects is important and will help us understand how cells deal with nutrient deprivation crisis. However, many of the important claims of this paper are either preliminary or not supported by the experiments reported. Indeed, important controls and statistics are missing. Moreover, I am not convinced that CLUH-Granules are different entity then Stress Granules (SGs). The fact that CLUH is not found in arsenite-induced SGs is not enough to support that claim. Much more experiments are needed (see below). The fact that G3BP1, one of the main promoters of SGs assembly, is also recruited to CLUH-granules raises the possibility of a link between SGs and these entities. The role of G3BP1 in CLUH formation and function should also be investigated to rule out any connection to SGs.

Other major issues:

- Figures 1 and 2: To correctly interpret the colocalization observations between CLUH and target mRNAs, a negative mRNA control is required. It is important to use an mRNA that is as abundant as *Hdha* and *Pcx* transcripts but does not associate with CLUH. Indeed, the percentage of colocalization between the negative control mRNA and CLUH will set the background of the experiment.

- Figure 1:

- o Here the authors conclude that starvation triggers the formation of CLUH granules that in turn recruit CLUH mRNA targets such as *Pcx* and *Hdha*. Since under basal conditions these granules are also detected, it is important to quantify the starvation-induced increase. How many more CLUH granules under HBSS conditions compared to control?
- o The authors should provide statistics for all the graphs shown. Is the increase in the co-localization of an mRNA with CLUH under HBSS condition significant?
- o Here the authors conclude that CLUH granules are different from SGs. While based on their observation this is possible, more experiments are needed to support this claim. Are CLUH granules sensitive to SG inhibitors such as Cycloheximide? Do these granules form in cells depleted or overexpressing G3BP1?
- o In these experiments, the authors should demonstrate that there is no bleed-through between channels. This is important since the data show that all CLUH granules have mRNAs.

- Figure 2:

- o The authors used the ribopuromycylation assay to investigate the translation status of mRNAs recruited into CLUH-granules.
- o As for Figure 1, statistics are needed. Quantification of the increase in CLUH granules and its significance should be provided.
- o Did the authors correct for the chromatic aberration that usually happens in these ribopuromycylation experiments.

- Figure 2G:

- o As mentioned above, the conclusion that CLUH-granules are distinct from SGs is premature. The possibility exists that these CLUH-Granules are SGs that require CLUH under starvation conditions. It is indeed possible that the puromycin positive granules detected under starvation conditions are simply a reflection of an mRNA that undergone the initial steps of translation and was just stalled and trapped in these entities in response to starvation. Based on the data and conclusions shown in

Figure 3, these granules are also able to inhibit the translation of some of the mRNAs they recruit. Did the authors validate such an observation?

o The fact that in response to arsenite CLUH does not localize to G3BP1 containing SGs, does not provide enough support, on its own, to conclude that CLUH-granules are distinct from SGs. Indeed, it is possible that arsenite excludes CLUH from SGs. Since it is well-known that the overexpression of G3BP1 by itself induces SGs formation in the absence of any stress, the authors should assess whether G3BP1 expression is modulated by CLUH and/or starvation. Also, the experiments shown in Figure S3 should be repeated in hepatocyte cells, and G3BP1 expression levels should be determined in WT and CLUH ^{-/-} hepatocytes.

- Figure 3: The authors should validate some of the identified transcripts using qRT-PCR and proteins by western blot. Additionally, as per the abstract statement, one of the authors' main conclusion is that CLUH-granules protects from decay many mRNAs known to encode factors involved in mitochondrial energy-converting pathways. No experiments are provided to support this claim. In their JCB 2017 paper (Schatton et al., JCB 2017) the authors demonstrated that CLUH protects many of these mRNAs from decay. However, in the actual study, the claim is that CLUH do so via CLUH-granules. Hence, it is important to show that this stabilization effect of CLUH on its target messages under starvation conditions occurs specifically via the assembly of CLUH-granules. Does G3BP1 play a role in this effect?

- Figure 4: The authors assess the impact of CLUH-granules on the mTOTC1 pathway in response to starvation. The authors chose to follow the formation of the CLUH-granules by probing for G3BP1. As mentioned above, it is possible that starvation/CLUH up-regulates G3BP1 expression, which in turn assembles these granules. It is important that the role of G3BP1 in mediating or not these outcomes is determined. Some of the experiments of Figure 4 could be repeated in the absence of G3BP1, which should mimic the results obtained in CLUH ^{-/-} cells.

- Figure 5: Can the authors observe the same beneficial effects of rapamycin in the absence of GBP1 in cells expressing or not CLUH?

- Figure 6 A-E: Statistics needs to be provided. Are the observed differences significant?

1st Revision - authors' response

27th December 2019

RESPONSE TO THE REFEREES' COMMENTS

Referee #1

MAJOR POINTS

1. In their colocalization studies (Fig. 1E,F; Fig. 2B), the authors make use of Mander's co-efficients; the spread of coefficients in the figures is wide - can the authors provide statistical analysis to assess whether the changes they see are actually statistically significant?

We have added statistical analysis to the data in Figs. 1 and 2. The changes are significant.

2. It seems a bit confusing that Pcx and Hadha associate with granules more upon starvation, but in the presence of glucose the granules do exist and the mRNAs seem to be translated. Are these mRNAs truly translated when associated with these granules (which would contrast with the situation for P-bodies and stress granules, in which mRNAs are not actively translated)? I.e., do puromycin data speak to the actual synthesis of these proteins within the granules or to other components of the complexes? Given that puromycin labeling is used, these cannot presumably be complete proteins, so that should rule out that they have been made elsewhere and subsequently joined the CLUH granules.

To answer the Referee's questions, we performed additional controls for the ribopuromycylation experiments. First, we checked that the puromycin signal was specific and not present when puromycin was not added (Appendix Fig. S1). Then, we confirmed that the signal decreased when incubating cells with arsenite that induce the formation of SGs (Fig. 2 A, B). Lastly, we used homoharringtonine (HHT), which blocks the first round of translational elongation. Such an inhibitor would dissociate polyribosomes actively synthesizing proteins due to ribosome run off. Consistently, preincubation with HHT, reduced the puromycin signal and the number of puromycin granules in HBSS (Fig. 2A-C). HHT also prevented the detection of puromycin granules containing CLUH and the mRNAs that we investigated (Appendix Fig. S3). Together, these results indicate that at least a subset of CLUH granules are translationally active. However, we also noted that some of the puromycin granular signal observed in HBSS were resistant to HHT and still contained CLUH (Fig. 2D-F). One possible explanation is that mRNAs in these granules are stalled at the level of elongation and/or termination, and are thus unaffected by HHT. However so far, we do not know which mRNAs are present in these granules. In conclusion, CLUH RNP particles appear heterogeneous and playing complex roles. Future studies will be required to fully clarify how CLUH function in the granules is regulated. We have further discussed our findings and the remaining open issues in the Discussion.

3. Bottom of p. 9 and Fig. 3B - "starvation-induced granules" - the authors same the number of granules +/- starvation, so the granules are not induced by starvation. They can only conclude that the granules containing G3BP1 mRNA are partly dependent on CLUH (some may be other kinds of granules, I guess).

In the old Fig. 3B (now Fig. 4G), we quantified the percentage of cells showing distinguishable granules, independently from the number of granules (this is specified in the figure legend). The difference between basal and HBSS conditions is in the number of granules that are seen (see also Fig 1). However, quantification of the number of granules in an automatic unbiased manner proved to be very complicated, owing to the fact that there is a lot of G3BP1 signal outside granules. The Referee is right: we cannot exclude that we are also detecting other kinds of G3BP1-positive granules.

4. Bottom of p. 9 and Fig. S3A: the authors cannot conclude that CLUH induces granules without data for control cells (to see if there are MORE granules in the cells that overexpress CLUH. It is not really clear what Fig. S3B shows - only really shows that ectopically overexpressed FLAG-CLUH associates with granules, which is not surprising, as endogenous CLUH does this).

We have repeated these experiments by overexpressing an untagged version of CLUH and we have compared cells overexpressing CLUH to cells with endogenous CLUH levels. We show that overexpression of CLUH leads to granule formation in absence of any other stress in about 40% of cells (Fig. 4 A-C).

5. P. 10, the opening statement is very far-reaching - what is the justification for this? ('organize the cellular metabolic response to nutrient availability').

We have reformulated the opening statement to: "To further understand the physiological role of CLUH granules..."

6. Related to Fig. 3: the use of the terms 'upregulated' and 'downregulated' is misleading as it implies an active process. Presumably, loss of CLUH causes destabilisation of some mRNAs (cf. their earlier paper, Schatton et al., ref. 18), for example, in which case the authors should just say 'was decreased'.

We have reformulated the text by avoiding the terms upregulated and downregulated.

7. Fig. S4B - some of the changes they report are small, e.g., *antilog2* 0.25, which is <1.2 fold change - what can they conclude about the biological significance of such small significance; validation would help.

The displayed GO terms are based on a 1-dimensional annotation enrichment (Cox and Mann. *BMC Bioinformatics*. 2012;13 S12. doi:10.1186/1471-2105-13-S16-S12). Since this summarizes the fold changes of all proteins annotated with a specific term, even small shifts in pathways can be identified. Only statistically different annotations that are above the Benjamini-Hochberg FDR cut off of 0.02 are displayed and listed (Appendix Dataset S3). Due to the short starvation period of only 2 hours the observed changes probably indicate early proteome remodeling in response to nutrient depletion.

8. Fig. 3G - needs better information on which mitochondrial enzymes changed, and the average extent. Fig. 3F does not show what happens to mitochondrial R-proteins in general (only four are pinpointed), although they are mentioned in the text; why? Also need to identify or give examples of 'mitochondrial enzymes involved in catabolic pathways' (just giving gene names is inadequate).

Fig. 3F (now Fig. 5E) shows pathways and not individual proteins. We have carefully revised the text to make it clear when we consider pathways or individual proteins. We have also been more specific when we discuss mitochondrial catabolic pathways. In addition, in Figure 5F, we have enlarged the part of the graph showing proteins encoded by CLUH targets that are unaffected or slightly upregulated in absence of CLUH. Annotation of these proteins shows that they largely correspond to mitochondrial ribosomal proteins.

9. Page 13 and Fig. 4: they don't mean they saw 'an increase in phosphorylation of 4E-BP1 and PRAS40 upon 24 h starvation'; their phosphorylation actually falls, but less so in the CLUH-KO cells. This part is confusingly written - there is no hyperactivation of mTORC1, it is just less sensitive to inhibition by nutrient starvation.

We have reformulated the text to avoid misunderstandings.

10. In Fig. 3G,H - why do the authors use Cos-7 siRNA cells, rather than cells from the KO mice?

We used COS7 to confirm data upon downregulation to exclude any compensatory effect in the knock-out mice. For us, it was important to be sure that data were reproducible in a completely different system. However, we have removed data showing mTORC1 signaling in COS7 from the revised manuscript to make the story simpler.

11. Fig. 4A and B: The authors need to show G3BP1-mTOR/RAPTOR co-localization under basal conditions. Also, for these and other instances of imaging

figures), it'd be helpful - and much more convincing - to show a population of cells (more than 5) rather than just showing one single cell.

We have now included in Fig. 6K colocalization of G3BP1 and mTOR also under basal condition and included a quantification showing that colocalization increases upon HBSS (Fig. 6L). In this case, we prefer to show only one representative cell. Hepatocytes are very large cells and showing more cells in one panel would make the relevant data hard to see.

12. On page 13, the authors claim that "Furthermore, in basal conditions the granules were next to LAMP1-positive structures, where mTORC1 is active, while they colocalized with LAMP1 in HBSS (Fig. 4B)." However, as far as I can see from Fig. 4B, G3BP1 is pretty much dispersed everywhere within the cell and not just localized with LAMP1. The authors really need to show some enlarged areas (like in Fig. 2A) and present the data with stats to support their claim (if true) here.

We have added a new experiment (Fig. EV3) where we evaluate co-localization of G3BP1 and LAMP1 in basal conditions and after 30 min or 2 h of starvation in WT and KO hepatocytes. For this experiment, only cells showing clear granules were selected and Manders' coefficient was calculated. We observed that KO hepatocytes with granules show colocalization with LAMP1 already at 30 min, while in WT cells the co-localization is maximal at 2 h. These data support the hypothesis that KO cells show less G3BP1 granules because they are subjected to a faster turnover. Consistently, the levels of G3BP1 are lower in KO hepatocytes (Fig. 4H, I). Moreover, we now show that the autophagic flux is increased in absence of CLUH (Fig. 7H-J, see also answer to point 20).

13. In Fig. 4C and E, the authors need to show the phosphorylation status of some other conventional mTORC1 downstream targets such as S6K1 or rpS6, like they did in G and H. I can understand P-S6K1 is difficult to look at in primary cells, but at least P-rpS6 should be an easier one; also very important to quantify P-protein/total protein for each one across at least three experiments (e.g., for 4E-BP1). The same applies to Fig. 4G/H. This is crucial as the claimed effects are modest/subtle. The blot for 4E-BP1 pT37/46 is of poor quality (Minor point: actually the residue numbers are T36/T45 in mice). In the title for legends of Fig. 4, "regulate" sounds quite vague, better change to a more precise word like "contributes to" or "prevents".

We have examined also the phosphorylation of RPS6 and quantified the ratio of pRPS6 and p4E-BP1 to the corresponding non-phosphorylated forms both in hepatocytes and livers (Fig 6A-F). We have also increased the number of mice in which we have performed these analyses. The increase in p4E-BP1 is more consistent than that of pRPS6 (not significant owing to large variability, but in general increased in KO). This is interesting, since the regulation of translation of mitochondrial proteins by mTORC1 is 4E-BP dependent (Morita et al. 2013). We have corrected the wrong labeling of the figure.

14. Regarding the co-localisation (or adjacency) of CLUH with LAMP1 - given that the authors show effect on signaling downstream of mTORC1, which is activated on the lysosomal surface: is the lysosomal association of mTORC1 altered in CLUH-KO vs. WT cells?

Unfortunately, we could not directly test this since the antibodies against LAMP1 and mTOR are both polyclonal.

15. Fig. S5E - what are the different bars in the graphs? This is not explained in the legend.

We have corrected the figure and the respective figure legend (now Fig. 8J, L).

16. The observation that rapamycin restores mitophagy in CLUH-deficient cells does not, as claimed, prove that the effect of CLUH is via mTORC1 - it just shows that the effect of lack of CLUH is overcome by rapamycin, a drug known to promote autophagy through mTORC1 inhibition.

The Referee is correct and we have now rephrased this in the Result section.

17. Fig. 5K: "Instead, mitochondrial proteins encoded by CLUH target mRNAs were decreased in purified mitochondria (Fig. 5K)." This can be visualised in the COX IV blot, although this is less convincing with respect to NDUFA9, and not at all with ATP5a. Therefore, I am not sure the general statement the authors make is a valid one.

We have repeated the western blot by including the protein products of the CLUH targets that we analyze in this study for consistency and we now show in Fig. 8C, D that they are all decreased.

18. "In further agreement with a mitophagy block, LC3 signal co-localized more with mitochondria in liver tissue lacking CLUH compared to controls (Fig. 5L, M)." The graph of one of the Manders' coefficients in 5M definitely indicates this, but the two images in 5L do not look much different from each other.

We have changed the image to better reflect the data (Fig. 8E). We also show a larger field and more enlarged areas.

19. "An accumulation of bigger autophagosomes was confirmed in primary hepatocytes derived from Li-CluhKO mice (Fig. 6A-C). Furthermore, also in cells more LC3-positive structures co-localized with mitochondria, and rapamycin treatment was effective in restoring the co-localization between mitochondria and LC3 puncta to WT levels (Fig. 6D, E)." Again, with nonetheless much variation, the graphs (6B, C and E) indicate this, but it is difficult to discern much difference between each of the three pictures in 6D (in particular WT v.s KO vs. KO + rapamycin, representative of 6E).

We have changed the image to better reflect the data (Fig. 8G). We show a larger field and more enlarged areas.

20. One item that would strengthen the 'story' is the classic without and with CQ/Bafilomycin assay, with/without CLUH expression, followed by immunoblotting for 62 and LC3. This could be inserted into Figure 5. This would confirm that there is a block in autophagic flux, since several of the other results could arise from increased synthesis rather than a block in degradation.

We thank the Referee for this suggestion. We have performed this experiment in hepatocytes and have revealed that the autophagic flux is not impaired but even enhanced in basal conditions upon CLUH KO (Fig. 7H-J). This was an extremely interesting finding because it showed that CLUH inhibits bulk autophagy but promotes mitophagy. Since lack of CLUH recapitulates the effects on autophagy

and mitophagy driven by Bre5, the yeast orthologue of G3BPs, we postulated that recruitment of G3BPs to CLUH granules would limit their activity. We have therefore tested if downregulation of G3BP1 and G3BP2 rescued the mitochondrial clustering caused by CLUH downregulation in Cos7 cells. In Fig. 9E-G, we show that this is indeed the case.

MINOR POINTS:

21. *What does CLUH-starved hepatocytes mean (p. 10, last line)?*

Sorry, we have corrected this sentence.

22. *Minor point: p. 15 - torin 1 is an mTOR kinase inhibitor rather than an irreversible inhibitor. It works differently from rapamycin.*

We have corrected this point.

Referee #2

The article of Pla-Martin et al joins the series of beautiful papers from Rugarli-lab in identifying and characterising CLUH as an RNA-binding protein for nuclear-encoded mRNAs of proteins to be translocated to mitochondria, and as a major regulator of metabolism, tightly communicating with mTOR complex to fine-tune mitochondrial and cytoplasmic translation in variable nutrient situations.

The authors have previously shown that CLUH binds a subset of nuclear-encoded transcripts and promotes their stability and translational output involved in OXPHOS, TCA cycle, amino acid degradation as well as ketogenesis. Here they show that the membraneless organelle-like G3BP1-positive RNA-granule-structures function in close vicinity to mTORC1 complex, and that CLUH is the driver of these RNA-granules. They propose that CLUH has a complex interplay with mTORC1, suppressing mTORC1 from overactivity at the time of early starvation. Utilizing hepatocyte cultures and mouse liverKO models, the authors report that nutrient state affects mRNA binding to the CLUH granules and that the granules are not only for mRNA storage, but translationally active. Furthermore, their RNAseq experiments of KO and WT cells in starved and fed conditions suggest that CLUH promotes mitochondrial catabolic functions, but inhibits mitochondrial translation.

The paper is well-written and the results are of high quality and provide novel information of coordination of translation in starved and fed state. The supplementary Tables did not exist in the version that I got for review.

We thank the Referee for these positive comments.

Comments:

The proteomics-data suggest that under starved conditions, CLUH-KO cells upregulate mitochondrial translation-associated proteins. As this does not yet indicate activation of translation, which by itself would be somewhat surprising, I

would like to see a mitochondrial translation assay, to test whether the translational activity is induced. The potential function of CLUH as a cytoplasmic regulator of mitochondrial ribosomal function would be very interesting.

We have performed an *in organello* translation assay using WT and KO hepatocytes. We do not see a translational inhibition, as it has been previously reported in CLUH KO HeLa cells (growing in glucose-rich medium). It should be noted however that primary hepatocytes show very limited mitochondrial translation. The blots were exposed for 4 weeks. For this reason, we prefer not to include this blot in the manuscript, but we show it here below for clarification.

The experiments of auto/mitophagy do not allow the conclusions made of CLUH participating in regulator of mitophagy regulation together with mTORC1. I do agree that that CLUH-KOs seem to have stalled autophagic flux (p62 accumulation, LC3 puncta) and this is resolved by rapamycin. However, this result is not necessarily an indication of a CLUH-mTORC1 functional link, because the lack of CLUH causes deficiency of oxphos enzymes (as visible in fig 4K, for oxphos complexes CI, IV and V), presumably because of lack of nuclear-encoded oxphos subunits. mTORC1 has been shown to activate upon oxphos deficiency, and rapamycin ameliorates those phenotypes (e.g. Kaeberlein lab) probably by allowing auto/mitophagy. Therefore, roles of CLUH for mitophagy are likely to be secondary. Induction of autophagy may also explain the improved viability that they see in rapamycin treated CLUH-KOs. The authors should revise the text to be less definitive on the mechanistic link of CLUH in mitophagy/autophagy caused by mitochondrial dysfunction.

We also have considered the possibility that mTORC1 hyperactivation in absence of CLUH is the result of respiratory deficiency. However, we do not see any evidence for activation of downstream transcriptional responses (for instance Atf4, Atf5, or genes involved in folate or serine metabolism) in hepatocytes or the liver. We have now added these data in Appendix Fig S5. Thus, we favor the hypothesis that CLUH limits mTORC1 activation by recruiting mTOR into the granules. However, we agree that rapamycin experiment per se does not prove that the mitophagy block is caused by mTORC1 hyperactivation. We have rephrased the text carefully to avoid a strong conclusion in this respect. In addition, we now show (Fig. 9E-G) that also downregulation of G3BPs prevents the clustering upon CLUH

downregulation, so it is clear that hyperactivation of mTORC1 is not the only culprit for this phenotype.

For apoptosis induction it would be good to see Western blots of starved mouse livers and cell lines, to get an overall view of the extent of caspase-3 cleavage.

The extent of apoptosis is minor *in vivo*, and we could not detect signal on a blot for cleaved caspase3. This is why we have counted cells in immunofluorescence/immunohistochemistry.

Minor comments: the abstract has a lot of abbreviations not clear for experts of this topic. They should be explained and written open.

We have rewritten the Abstract and tried to make it clear also for non-experts in the topic.

Referee #3

In this paper, the authors investigate the mechanisms by which the RNA Binding protein CLUH (clustered mitochondria homolog) in modulating the expression of mitochondrial genes and its role during nutrient fluctuation. In response to nutrient, starvation mitochondria change its gene expression profile to adapt to this stressful condition and ensure energy production and cell survival. In previous work, that authors' lab demonstrated that CLUH regulates the expression of transcripts encoding key factors that affect mitochondria function and adaptation to starvation. In this work, the authors provide a series of observations suggesting that in response to nutrient starvation, CLUH form specific cytoplasmic RNA granules, via which it modulates mitochondria gene expression and fate. They suggest that the CLUH-granules are different from Stress Granules, even though they recruit G3BP1 and TIA-1, two bonafide SG promoters. In the situation where nutrients are available and abundant, mTORC1 plays a key role in promoting the expression of factors involved in oxidative phosphorylation and activation of mitochondrial translation, the expression of many of which is regulated by CLUH. The authors provide experiments showing that CLUH, possibly via its granules, interferes with the premature activation of the mTORC1 pathway in response to nutrient starvation, leading to the inhibition of mitochondria anabolic pathways. Indeed, depleting CLUH, which in turn prevents the formation of CLUH-granules, triggers the hyperactivation of mTORC1 leading to cell death.

In their JCB 2017 paper, the authors demonstrated the importance of CLUH proteins in regulating mitochondrial function and adaptive capacity to help the cell deal with stressful conditions such as nutrient starvation. Hence, delineating the mechanisms behind the CLUH-mediated effects is important and will help us understand how cells deal with nutrient deprivation crisis. However, many of the important claims of this paper are either preliminary or not supported by the experiments reported. Indeed, important controls and statistics are missing. Moreover, I am not convinced that CLUH-Granules are different entity then Stress Granules (SGs). The fact that CLUH is not found in arsenite-induced SGs is not enough to support that claim. Much more experiments are needed (see below). The fact that G3BP1, one of the main promoters of SGs assembly, is also recruited to CLUH-granules raises the possibility of a link between SGs and these entities. The role of G3BP1 in CLUH formation and function should also be investigated to rule out any connection to SGs.

We thank the Referee for their comments which were very helpful for improving our study. We have performed additional experiments to established the nature of the CLUH granules and their relationship with G3BP1. Moreover, we have added a number of controls and indicated the results of statistical tests. We believe that these new experiments (see below point-by-point response) answer most of the Referee' concerns.

Other major issues:

- Figures 1 and 2: To correctly interpret the colocalization observations between CLUH and target mRNAs, a negative mRNA control is required. It is important to use an mRNA that is as abundant as HdhA and Pcx transcripts but does not associate with CLUH. Indeed, the percentage of colocalization between the negative control mRNA and CLUH will set the background of the experiment.

We thank the Referee for bringing up this important point. We have now used as negative control the mRNA for *Actb*. We show that there is the same amount of colocalization (Manders' coefficient) between CLUH and *Actb* or *Pcx* or *Hadha* mRNAs in basal conditions, while in HBSS only the colocalization of CLUH with *Pcx* and *Hadha* significantly increases (Fig. 1A-F). It should be noted that the colocalization coefficient here considers the whole picture and not only the granules. Thus, the Referee was right, the co-localization with *Actb* gives us the measure of background signal observed in basal conditions. Indeed, in basal conditions we could see the mRNA of *Pcx* and *Hadha* co-localizing only in the granules. Since these are really few, they do not impact on the Manders' coefficient that considers the whole cell.

- Figure 1:

o Here the authors conclude that starvation triggers the formation of CLUH granules that in turn recruit CLUH mRNA targets such as Pcx and Hadha. Since under basal conditions these granules are also detected, it is important to quantify the starvation-induced increase. How many more CLUH granules under HBSS conditions compared to control?

We understand the Referee's comment. We discussed how to perform this quantification, but it is very difficult to do it in an unbiased automatic manner because of the fact that CLUH staining shows small puncta or bigger granules and some cytosolic signal. However, in HBSS bigger CLUH granules are more evident. We are not surprised that some CLUH granules are detected also under basal conditions. In fact, we see that these granules contain different translationally active mRNAs in different conditions.

o The authors should provide statistics for all the graphs shown. Is the increase in the co-localization of an mRNA with CLUH under HBSS condition significant?

We have added statistics for all graphs. The increase of the co-localization of CLUH with *Pcx* and *Hadha* under HBSS is significant (Fig. 1D, E).

o Here the authors conclude that CLUH granules are different from SGs. While based on their observation this is possible, more experiments are needed to support this claim. Are CLUH granules sensitive to SG inhibitors such as

Cycloheximide? Do these granules form in cells depleted or overexpressing G3BP1?

To answer these questions, we have performed a series of additional experiments. We have employed HeLa cells, because of the technical difficulty of downregulating G3BP1 in primary hepatocytes.

First, we tested if depletion of G3BP1 and the homologous gene G3BP2 impaired the formation of granules upon CLUH overexpression (Fig. 4 A-C). CLUH overexpression led to the formation of granules positive for G3BP1 and G3BP2, in absence of any stress and also in presence of cycloheximide (Fig. EV2E-F). Moreover, when we downregulated G3BP1 or both G3BP1 and 2 the granules were still formed, indicating that G3BPs are not necessary for CLUH to form granules (Fig. 4A-C). Second, we used live-imaging to follow G3BP1-positive granule formation in HBSS in WT and CLUH KO HeLa cells. We selected for imaging cells that did not show granules at the beginning of imaging, to avoid imaging of SGs that can be formed simply by G3BP1 overexpression. We found that G3BP1 forms granules in WT cells even in presence of cycloheximide, and these were not formed in KO cells (Fig. 4D, E). In contrast, both WT and KO HeLa cells formed G3BP1-positive granules upon arsenite treatment that were resolved by cycloheximide, and are therefore *bona fide* SGs (Fig. EV2I, J).

Based on these data, we conclude that CLUH can form granules that are not SGs and are independent from G3PBs for their formation.

o In these experiments, the authors should demonstrate that there is no bleed-through between channels. This is important since the data show that all CLUH granules have mRNAs.

We have always controlled for bleed-through between channels with each antibody. Moreover, not all CLUH granules have mRNAs. For instance, CLUH granules resistant to HHT are not positive for mRNAs (see new Appendix Fig S3).

- Figure 2: o The authors used the ribopuromycylation assay to investigated the translation status of mRNAs recruited into CLUH-granules.

o As for Figure 1, statistics are needed. Quantification of the increase in CLUH granules and its significance should be provided.

We have added statistics and several additional controls for the ribopuromycylation experiments shown in Fig 2.

o Did the authors correct for the chromatic aberration that usually happens in these ribopuromycylation experiments.

We are performing these experiments in the CECAD imaging facility where all microscopes are routinely tested for chromatic aberration using beads (see below).

- Figure 2G:

o As mentioned above, the conclusion that CLUH-granules are distinct from SGs is premature. The possibility exists that these CLUH-Granules are SGs that require CLUH under starvation conditions. It is indeed possible that the puromycin positive granules detected under starvation conditions are simply a reflection of an mRNA that undergone the initial steps of translation and was just stalled and trapped in these entities in response to starvation.

We performed additional controls for the ribopuromylation experiments. First, we checked that the puromycin signal was specific and not present when puromycin was not added (Appendix Fig. S1A). Then, we confirmed that the signal decreased when incubating cells with arsenite that induce the formation of SGs (Fig. 2 A, B). Then, we used homoharringtonine (HHT), which blocks the first round of translational elongation. Such an inhibitor would dissociate polyribosomes actively synthesizing proteins due to ribosome runoff. Consistently, preincubation with HHT, reduced the puromycin signal and the number of puromycin granules in HBSS (Fig. 2A-C). HHT also prevented the detection of puromycin granules containing CLUH and the mRNAs that we investigated (Appendix Fig. S3). Together, this indicates that at least some CLUH granules are translationally active. However, we also noted that some of the puromycin granular signal observed in HBSS was resistant to HHT and still contained CLUH (Fig. 2D-F). One possible explanation is that these granules are stalled at the level of elongation and/or termination, and are thus unaffected by HHT. We have further discussed these findings in the Discussion.

Based on the data and conclusions shown in Figure 3, these granules are also able to inhibit the translation of some of the mRNAs they recruit. Did the authors validate such an observation?

This is an exciting possibility that we address in the Discussion. Unfortunately, it is hard to validate this observation, without purifying the different types of CLUH granules to analyze their mRNA complement. This is definitively an important issue to be resolved in a future study.

o The fact that in response to arsenite CLUH does not localize to G3BP1 containing SGs, does not provide enough support, on its own, to conclude that

CLUH-granules are distinct from SGs. Indeed, it is possible that arsenite excludes CULH from SGs. Since it is well-known that the overexpression of G3BP1 by itself induces SGs formation in the absence of any stress, the authors should assess whether G3BP1 expression is modulated by CULH and/or starvation. Also, the experiments shown in Figure S3 should be repeated in hepatocyte cells, and G3BP1 expression levels should be determined in WT and CLUH -/- hepatocytes.

We checked if G3BP1 expression is modulated by CLUH. Indeed, in basal conditions, hepatocytes lacking CLUH have reduced G3BP1 levels (Fig. 4H, I). Upon starvation, however, the levels of G3BP1 increase in both WT and KO hepatocytes and are not different between the two genotypes, so it is unlikely that G3BP1 levels underlie our results. In addition, we included new data showing that bulk autophagy is increased in basal conditions when CLUH is not present (Fig. 7H-J). Together these data suggested that KO cells may show less G3BP1 granules at 2h HBSS because they are removed by autophagy. In line with this hypothesis, we found that at 30 min of starvation KO hepatocytes contained G3BP1 granules colocalizing with Lamp1. Therefore, it is possible that G3BP1 granules are less frequently observed in KO cells because of a higher turnover rate. We have discussed these new finding in the revised Discussion.

- Figure 3: The authors should validate some of the identified transcripts using qRT-PCR and proteins by western blot.

We present transcriptomics and proteomics analysis and we believe that validating results via qRT-PCR or western blot would be redundant also with our previous findings (Schatton et al. JCB 2017). However, the Referee can see in Fig. 8C that the levels of HMGCS2, HADHA, and PCX are reduced in mitochondria lacking CLUH.

Additionally, as per the abstract statement, one of the authors' main conclusion is that CLUH-granules protects from decay many mRNAs known to encode factors involved in mitochondrial energy-converting pathways. No experiments are provided to support this claim. In their JCB 2017 paper (Schatton et al., JCB 2017) the authors demonstrated that CLUH protects many of these mRNAs from decay. However, in the actual study, the claim is that CLUH do so via CLUH-granules. Hence, it is important to show that this stabilization effect of CLUH on its target messages under starvation conditions occurs specifically via the assembly of CLUH-granules. Does G3BP1 play a role in this effect?

We understand the Referee's concern, however to formally rule out this possibility we should be able to express a mutant form of CLUH in KO cells which retain its function but does not form granules. We believe that answering this question is beyond the scope of the present manuscript.

- Figure 4: The authors assess the impact of CLUH-granules on the mTOTC1 pathway in response to starvation. The authors chose to follow the formation of the CLUH-granules by probing for G3BP1. As mentioned above, it is possible that starvation/CLUH up-regulates G3BP1 expression, which in turn assembles these granules. It is important that the role of G3BP1 in mediating or not these outcomes is determined. Some of the experiments of Figure 4 could be repeated in the absence of G3BP1, which should mimic the results obtained in CLUH -/- cells.

As shown in Fig. 4H lack of CLUH does not upregulate, but downregulates G3BP1. Moreover, in Fig 9E-G, we show that lack of G3BP1 does not recapitulate, but rather rescues one of the hallmarks of CLUH deficiency, mitochondrial clustering.

- *Figure 5: Can the authors observe the same beneficial effects of rapamycin in the absence of GBP1 in cells expressing or not CLUH?*

We thank the Referee for this question. We have now performed experiments in which we downregulated G3BP1, G3BP2 or both in COS 7 cells depleted for CLUH (Fig 9E-G). Indeed, depletion of the G3BPs prevented mitochondrial cluster formation, as observed for rapamycin. We have discussed these findings in relation to the known function of the yeast G3Bp orthologue, Bre5, which promotes general autophagy but inhibits mitophagy.

- *Figure 6 A-E: Statistics needs to be provided. Are the observed differences significant?*

We have added statistics to show significance of the data (now in Fig 7 and 8).

2nd Editorial Decision

31st January 2020

As you will see, referee#2 finds that all criticisms have been sufficiently addressed and recommend the manuscript for publication. Referee #3 feels that the role CLUH granules in protecting mRNAs from decay is not yet supported by the data and recommends you to tone down this statement in the abstract and the discussion. Reviewer #1 asks you to rephrase the description of HHT action and to better discuss the biological relevance of the results in Fig. S4B.

In addition to satisfy the remaining referees' requests, there are a few editorial issues concerning text and figures that I need you to address before we can officially accept the manuscript.

REFEREE REPORTS

Referee #1:

COMMENT FOR EDITOR: When describing changes made to the text, it would be very helpful to the reviewers for the authors to indicate (by giving page and line numbers) where such changes have been made. This greatly facilitates the re-review process.

General summary and opinion: this study reports interesting and novel data concerning the role of ribonucleoprotein particles containing CLUH in regulating the translation of mRNAs, in particular ones encoding mitochondrial proteins and also modulates autophagy.

Minor concerns that should be addressed and suggestions for improving the study - no extra experiments are required but the authors should address these two points:

1. When describing the action of HHT, the authors write '[it] inhibits translation elongation and causes ribosome run-off'. This is very confusing, as inhibiting elongation blocks run-off. The point is that HHT inhibits the early transition from initiation to elongation, without interfering with the progress of ribosomes that are already engaged in elongation. Text needs to be changed.
2. Related to earlier point 7 and Fig. S4B, I still feel that the authors should comment on the small size of the changes they observe, which as they indicate in their response likely reflects the fact that

changes to the proteome are small at this early time.

Referee #2:

The authors have satisfactorily responded to all my concerns. I have no further criticism.

Referee #3:

I thank the authors for their great effort and "almost" through answers to my previous critiques. As such the manuscript has improved and many of the conclusion are now better supported.

However, one of the main conclusions, as stated in the abstract (..... prevents the premature degradation of mRNAs encoding mitochondrial catabolic proteins and promotes mitochondrial turnover in the liver...) and the discussion, is still not supported by the data. If the authors insist on the possibility that CLUH-granules are indeed involved in protecting the recruited mRNA from decay, this should be shown experimentally. I totally understand the authors' answer and the technical difficulty of such experiments. Therefore, it is premature to state that these granules protect mRNA from premature degradation and this conclusion should be down played or removed.

2nd Revision - authors' response

5th February 2020

Point-by-point response to the Reviewers

Referee #1

Minor concerns that should be addressed and suggestions for improving the study - no extra experiments are required but the authors should address these two points:

1. When describing the action of HHT, the authors write '[it] inhibits translation elongation and causes ribosome run-off'. This is very confusing, as inhibiting elongation blocks run-off. The point is that HHT inhibits the early transition from initiation to elongation, without interfering with the progress of ribosomes that are already engaged in elongation. Text needs to be changed.

To clarify this point, we changed the text as follows:

Results, page 6: "As expected, this signal was suppressed by pre-incubation with both homoharringtonine (HHT) and arsenite. HHT causes ribosome stalling at the initiator codon, but leaves unaffected downstream ribosomes already engaged in elongation, while arsenite is a potent inducer of SGs (Fig 2A, B)."

Discussion, page 15: "In addition, we found that a subset of CLUH granules can incorporate puromycin despite inhibition of the transition between translational initiation and elongation by HHT, suggesting that they might contain translationally stalled mRNAs."

2. Related to earlier point 7 and Fig. S4B, I still feel that the authors should

comment on the small size of the changes they observe, which as they indicate in their response likely reflects the fact that changes to the proteome are small at this early time.

We edited the text in the Result section, page 9 as follows:

“To systematically detect groups of mRNAs that participate in similar pathways and to reveal early proteome changes after the short starvation period, we performed one-dimensional (1D) pathway enrichment of transcriptome and proteome data (Appendix Tables S3), which allows the identification of even small shifts in pathways by summarizing the fold changes of all proteins annotated with a specific term. [...] Starvation triggered a modest reshaping of the transcriptome and proteome in cells of both genotypes, however analysis of pathways enriched or depleted specifically in WT or KO cells revealed differences (Appendix Fig S4A, B and Table S3). “

Referee # 3

I thank the authors for their great effort and "almost" through answers to my previous critiques. As such the manuscript has improved and many of the conclusion are now better supported.

However, one of the main conclusions, as stated in the abstract (..... prevents the premature degradation of mRNAs encoding mitochondrial catabolic proteins and promotes mitochondrial turnover in the liver...) and the discussion, is still not supported by the data. If the authors insist on the possibility that CLUH-granules are indeed involved in protecting the recruited mRNA from decay, this should be shown experimentally. I totally understand the authors' answer and the technical difficulty of such experiments. Therefore, it is premature to state that these granules protect mRNA from premature degradation and this conclusion should be down played or removed.

We understand the Referee's point. Therefore, we have revised the Abstract as follows:

“Upon starvation, CLUH regulates translation of *Hmgcs2*, involved in ketogenesis, inhibits mTORC1 activation and mitochondrial anabolic pathways, and promotes mitochondrial turnover, thus allowing efficient reprogramming of metabolic function.”

For consistency, we have also modified the summary at the end of the Introduction, by removing reference to autophagy of mRNAs for mitochondrial proteins.

However, we feel that the discussion is the place where hypotheses to be tested in the future can be put forward and discussed. We have however

reformulated the text, to make it clear what is a hypothesis and what needs to still be experimentally validated.

The relevant part in the discussion now reads (page 16):

“We speculate that sorting of target mRNAs into CLUH granules has a crucial role not only for their translation, but also to protect them against premature decay and autophagic degradation. Consistently, pathways associated with mRNA decay and the lysosome were enriched in proteomics analysis of starved KO hepatocytes. Furthermore, G3BP1-positive granules colocalized with LAMP1-positive organelles upon HBSS starvation with faster dynamics in KO hepatocytes. Whether the decreased occurrence of G3BP1-positive granules in hepatocytes lacking CLUH is totally explained by their faster turnover by autophagy or is also caused by failure of nucleation of the granules is a question to be addressed in the future. Furthermore, it is still unclear if the role of CLUH in stabilizing target mRNAs is dependent on granule formation.”

Accepted

7th February 2020

I am pleased to inform you that your manuscript has been accepted for publication in the EMBO Journal.

Corresponding Author Name: Elena I. rugarli

Journal Submitted to: EMBO J

Manuscript Number: EMBOJ-2019-102731